# A natural variant of the essential host gene *MMS21* restricts the parasitic 2-micron plasmid in *Saccharomyces cerevisiae*

**Michelle Hays[1,2†], Janet M Young[2], Paula F Levan[2], Harmit S Malik[2,3]\***

[1]Molecular and Cellular Biology program, University of Washington, Seattle, United States; [2]Division of Basic Sciences & Fred Hutchinson Cancer Research Center, Seattle, United States; [3]Howard Hughes Medical Institute, Fred Hutchinson Cancer Research Center, Seattle, United States

**Abstract** Antagonistic coevolution with selfish genetic elements (SGEs) can drive evolution of host resistance. Here, we investigated host suppression of 2-micron (2μ) plasmids, multicopy nuclear parasites that have co-evolved with budding yeasts. We developed SCAMPR (Single-Cell Assay for Measuring Plasmid Retention) to measure copy number heterogeneity and 2μ plasmid loss in live cells. We identified three *S. cerevisiae* strains that lack endogenous 2μ plasmids and reproducibly inhibit mitotic plasmid stability. Focusing on the Y9 ragi strain, we determined that plasmid restriction is heritable and dominant. Using bulk segregant analysis, we identified a high-confidence Quantitative Trait Locus (QTL) with a single variant of *MMS21* associated with increased 2μ instability. *MMS21* encodes a SUMO E3 ligase and an essential component of the Smc5/6 complex, involved in sister chromatid cohesion, chromosome segregation, and DNA repair. Our analyses leverage natural variation to uncover a novel means by which budding yeasts can overcome highly successful genetic parasites.

**\*For correspondence:**
hsmalik@fhcrc.org

**Present address:** †Department of Genetics, Stanford University, Palo Alto, United States

**Competing interests:** The authors declare that no competing interests exist.

## Introduction

Host genomes are engaged in longstanding conflicts with myriad selfish genetic elements (SGEs, or genetic parasites) (***Burt and Trivers, 2008***; ***Dawkins, 1976***; ***McLaughlin and Malik, 2017***). SGEs propagate within an organism or population at the expense of host fitness (***Burt and Trivers, 2008***). Many SGEs, including viruses, selfish plasmids, and other pathogens, must coopt the host's cellular machinery for their own survival: to replicate their genomes, to transcribe and translate their proteins, and to ensure their proliferation by passage into new cells (***Burt and Trivers, 2008***; ***McLaughlin and Malik, 2017***). If a host variant arises that can suppress SGEs (host restriction), this variant will be favored by natural selection and can rise in frequency in a population. If resistance has no fitness cost, such variants will rapidly fix within host species. Even if these variants are slightly deleterious, such variants could be maintained in quasi-equilibrium in host species (***Koskella, 2018***; ***Meaden et al., 2015***; ***Kraaijeveld and Godfray, 1997***; ***Sheldon and Verhulst, 1996***).

Studies in diverse biological taxa have leveraged genetic mapping strategies to identify quantitative trait loci (QTL) for host resistance to parasites (***Kane and Golovkina, 2019***; ***Cogni et al., 2016***; ***Kelleher et al., 2018***; ***Duffy and Sivars-Becker, 2007***). Such studies have revealed that host populations are more likely to harbor variation in resistance to coevolved, rather than to recently introduced, parasites (***Duxbury et al., 2019***). Studying parasites in their native host context therefore maximizes opportunities to discover host resistance mechanisms. However, it is often difficult to study natural variation in resistance, because hosts and/or parasites are often intractable in the

laboratory. Budding yeast provides an ideal system to study host-SGE genetic conflicts, with abundant genetic tools, together with resources for comparative and population genetics. Yeast species harbor a variety of SGEs including retrotransposable elements, RNA viruses and 2-micron (2μ) plasmids (*Rowley, 2017*; *Wickner, 1996*; *Kelly et al., 2012*; *Nakayashiki et al., 2005*; *Krastanova et al., 2005*; *Bleykasten-Grosshans and Neuvéglise, 2011*). Yet, despite its long history as a popular model eukaryote, natural variation in cellular immunity factors against SGEs has been largely uncharacterized in *S. cerevisiae* and related species (*Rowley et al., 2016*; *Czaja et al., 2019*; *Scholes et al., 2001*; *Maxwell and Curcio, 2007*; *Rowley et al., 2018*). Here, we investigated whether *S. cerevisiae* strains harbor genetic variants that allow them to resist a highly successful SGE: 2μ plasmids.

2μ plasmids are nuclear SGEs found in multiple, divergent budding yeast species (*Blaisonneau et al., 1997*; *Utatsu et al., 1987*; *Chen et al., 1992*; *Peter et al., 2018*). They are best characterized in *S. cerevisiae*, where they are found in high copy numbers:~50 copies per haploid and ~100 copies per diploid cell (*Veit and Fangman, 1988*; *Zakian et al., 1979*). 2μ plasmids are stably transmitted through vertical inheritance. However, even if they are lost stochastically, they can be reintroduced via sex and transmitted via non-Mendelian inheritance through meiosis: even if only one haploid parent initially has 2μ plasmids, all four meiotic progeny typically receive plasmids (*Futcher and Cox, 1983*). Their widespread prevalence in *S. cerevisiae* and other budding yeast species has raised the question of whether 2μ plasmids might be more commensal than parasitic. In the mid-1980s, two seminal studies showed that *S. cerevisiae* strains carrying 2μ plasmids (*cir*⁺) grew 1–3% more slowly than did their *cir*⁰ counterparts under laboratory conditions; thus 2μ plasmids confer a clear fitness defect (*Futcher and Cox, 1983*; *Mead et al., 1986*). Recent studies have reinforced the fitness defect associated with carriage of 2μ plasmids (*Harrison et al., 2012*; *Harrison et al., 2014*). Furthermore, many mutant yeast strains, which are sick in the presence of 2μ plasmids, can be partially rescued when 'cured' of their 2μ plasmids (*Dobson et al., 2005*; *Zhao et al., 2004*). For example, *nib1* mutants (a hypomorphic allele of *ULP1* [*Dobson et al., 2005*]) form 'nibbled' colonies in the presence of 2μ plasmids due to colony sectoring from cells that stop dividing when overburdened with 2μ plasmids, but form smooth (wild-type) colonies in their absence (*Dobson et al., 2005*). These and other data (*Zhao et al., 2004*) suggest that 2μ plasmids impose a selective burden on yeast, both under rapid laboratory growth conditions as well as in times of stress. In contrast to bacterial plasmids, which can harbor host-beneficial 'cargo' genes, such as antibiotic resistance genes, no such beneficial genes have ever been observed in natural 2μ plasmids (*Bennett, 2008*). Indeed, there are no known conditions in which 2μ plasmids are beneficial to the host, further supporting that its presence is likely the result of efficient parasitism. Although they are stable in *S. cerevisiae*, experimental studies show that 2μ plasmids exhibit lower copy number and decreased stability when introduced into exogenous species (*Murray et al., 1988*). These findings suggest that 2μ plasmids have co-evolved with host genomes to become a successful genetic parasite of yeasts.

To be successful, 2μ plasmids must replicate and segregate with high fidelity into daughter cells during both yeast mitosis and meiosis. Without these capabilities, plasmids risk being lost from the population as their host cells are outcompeted by plasmid-less daughter cells. Yet, 2μ plasmids encode just four protein-coding genes (represented by arrows in *Figure 1A*) in *S. cerevisiae*. *REP1* and *REP2* encode plasmid-encoded DNA-binding proteins that bind to the 2μ *STB* locus to mediate segregation (*Jayaram et al., 1983*; *Velmurugan et al., 1998*; *Veit and Fangman, 1988*). Mutations in *REP1* and *REP2* significantly impair segregation fidelity, resulting in failure to transmit plasmid to daughter cells, and subsequent loss from the host population (*Murray and Szostak, 1983*). If copy number drops below a certain threshold within a host cell, 2μ plasmids activate an amplification mechanism that relies on plasmid-encoded *FLP1* (*Murray et al., 1987*; *Som et al., 1988*). *FLP1* encodes a recombinase that creates plasmid structural rearrangements during host S phase via the *FRT* sites, facilitating over-replication via rolling circle replication using host replication machinery (*Murray et al., 1987*; *Som et al., 1988*; *Zakian et al., 1979*; *Volkert and Broach, 1986*; *Dobson et al., 1988*). *RAF1* encodes a protein that regulates the switch to copy number amplification (*Murray et al., 1987*). Due to this minimal genome, 2μ plasmids rely on host factors for genome replication and segregation during host cell division (*Zakian et al., 1979*; *Rizvi et al., 2018*; *Prajapati et al., 2017*; *Ma et al., 2013*; *Sau et al., 2014*; *Ghosh et al., 2007*; *Sau et al., 2015*).

Previous studies have identified host factors required by the 2μ plasmid. For instance, in addition to DNA replication and origin licensing factors, 2μ plasmids require host factors to facilitate proper

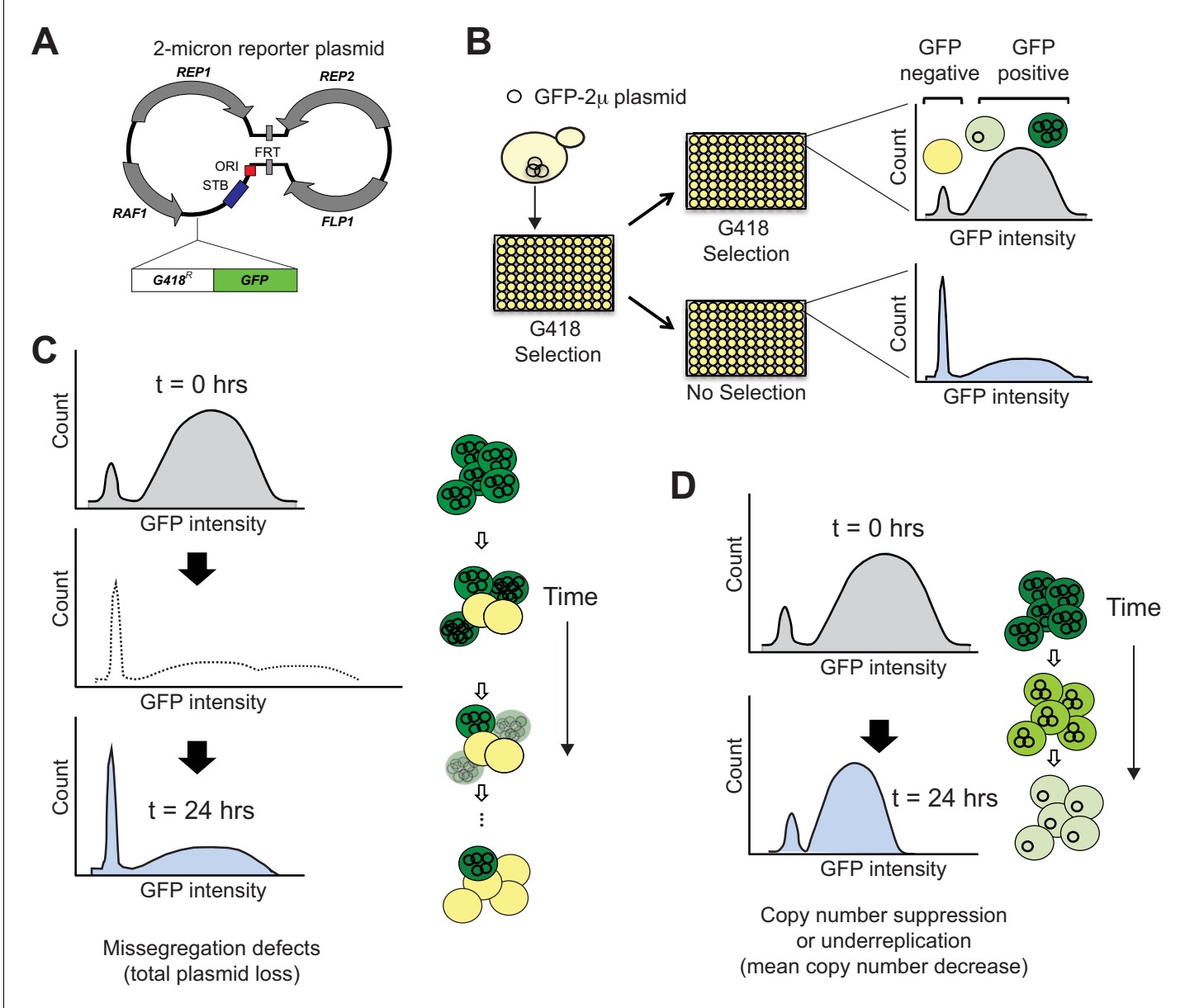

**Figure 1.** SCAMPR, a novel method to measure 2μ plasmid stability and dynamics. (A) Schematic of GFP-reporter 2μ plasmid. The endogenous 2μ plasmid encodes an origin of replication (ori), four protein-coding genes (*REP1*, *REP2*, *RAF1*, *FLP1*) and their interacting DNA loci (STB, and FRT). The GFP-2μ reporter plasmid described here utilizes the full 2μ genome with an additional integrated G418-resistance and GFP expression cassette. (B) A Single Cell Assay for Measuring Plasmid Retention (SCAMPR) utilizes the dual reporter cassette: G418 resistance to ensure plasmid retention while under selection and GFP to facilitate screening of plasmid-positive cells. Cells with the reporter plasmid are kept on G418 selection to ensure the plasmid is present at t = 0 and either released to media without selection or passaged with continued G418 selection. Comparing the GFP intensities of the cell populations with and without G418 selection after 24 hr reveals the plasmid retention dynamics and population heterogeneity of the host genetic background (*Figure 1—figure supplement 1*). SCAMPR can therefore distinguish between alternate mechanisms of plasmid instability, illustrated in (C) and (D), or the relative contribution of both mechanisms. (C) Gross segregation defects in which plasmids are not distributed to both daughter cells would cause an increase in GFP-negative cells, as well as an increase in 'super-green' cells that retain twice as many two plasmids (light shading, dotted histogram). However, we infer that these cells would either be lost or not proliferate due to growth defects associated with high plasmid copy number. As a result of this selection, we expect to see a rapid increase in GFP-negative cells but no dramatic change in the median expression of (surviving) GFP-positive cells. (D) Plasmid instability caused by under-replication or copy number suppression would not cause a precipitous decline in GFP-positive cells as in (C) but would instead lead to a reduction in the median GFP intensity of the GFP-positive cells. The online version of this article includes the following figure supplement(s) for figure 1:

**Figure supplement 1.** SCAMPR analysis for permissive and non-permissive *S. cerevisiae* strains.

partitioning into daughter cells, including many spindle-associated proteins (*Zakian et al., 1979*; *Rizvi et al., 2018*; *Prajapati et al., 2017*; *Ma et al., 2013*; *Sau et al., 2014*; *Ghosh et al., 2007*). Furthermore, host-mediated post-translational SUMO-modification of plasmid-encoded proteins has been shown to have a profound effect on 2μ plasmid stability and host fitness. For example, failure to sumoylate the Rep proteins impairs plasmid stability, whereas deficient sumoylation of Flp1 recombinase leads to recombinase overstabilization, resulting in massively increased plasmid copy number and extreme reduction in host cell fitness (*Zhao et al., 2004*; *Burgess et al., 2007*; *Pinder et al., 2013*). Indeed, mutations in SUMO E3 ligases *Siz1, Siz2*, the SUMO maturase *Ulp1*, and the SUMO-targeted ubiquitin ligase *Slx8* all lead to hyper-amplification and host cell defects (*Zhao et al., 2004*; *Burgess et al., 2007*; *Pinder et al., 2013*). These host-plasmid interactions provide potential means for the host to curb deleterious proliferation of plasmids. However, it is unclear whether gain-of-function alleles exist that restrict or eradicate plasmids.

Until recently, 2μ plasmids have been largely omitted from studies of genetic variation in yeast. Although prior work has predominantly focused on canonical A-type 2μ plasmids (found in laboratory *S. cerevisiae* strains), recent studies revealed that 2μ plasmids are quite diverse in budding yeast populations (*Peter et al., 2018*; *Strope et al., 2015*). These analyses identified C-type plasmids, extremely diverged D-type plasmids and a 2μ plasmid introgression into *S. cerevisiae* from the closely-related species *S. paradoxus* (*Peter et al., 2018*; *Strope et al., 2015*). Moreover, previously identified B- and newly described B*-type plasmids were shown to be a result of recombination between A and C types (*Peter et al., 2018*; *Strope et al., 2015*; *Xiao et al., 1991a*; *Xiao et al., 1991b*). Furthermore, these studies revealed that there are multiple, distinct strains of *S. cerevisiae* that do not harbor any 2μ plasmids. Yet, it remains unknown whether 2μ plasmid absence in these strains is the result of stochastic loss or an inherent host trait.

Host cells could influence 2μ plasmid fitness by affecting their copy number, stability, or population heterogeneity. However, these parameters are not captured in traditional plasmid loss assays, which measure either plasmid copy number averaged across the entire population, or plasmid presence versus absence independent of copy number. To quantitatively capture all of these parameters, we developed a new high-throughput, single-cell, plasmid retention assay, SCAMPR (Single-Cell Assay for Measuring Plasmid Retention). We identified three yeast strains that naturally lack 2μ plasmids and reproducibly show a high rate of mitotic instability of 2μ plasmids upon plasmid reintroduction. Focusing on one resistant strain, we used SCAMPR to show that resistance is a dominant, multigenic trait. Using QTL mapping by bulk segregant analysis, we identified one significant genomic locus that impairs 2μ mitotic stability. A candidate gene approach within this locus showed that a single amino acid change in *MMS21* contributes to plasmid instability. *MMS21* is a highly conserved E3 SUMO ligase and an essential component of the Smc5/6 complex, which has not previously been implicated in 2μ biology. Thus, our study reveals a novel pathway by which 2μ resistance has arisen and persists in natural populations of *S. cerevisiae*.

## Results

### SCAMPR: Single-Cell Assay for Measuring Plasmid Retention

To determine if there is heritable natural variation in 2μ plasmid stability in *S. cerevisiae* strains, we needed an assay to measure plasmid maintenance at the single-cell level. Traditionally, plasmid loss dynamics have been measured by two types of assays. The first of these is the Minichromosome Maintenance (MCM) assay, in which strains containing plasmids with selectable markers are assessed for plasmid presence versus absence by counting colonies on both selective and non-selective media over time (*Maine et al., 1984*). Due to the labor intensiveness of the assay, MCM is low-throughput since different dilutions need to be tested to recover and reliably count 30–300 colonies per plate. Furthermore, as only a single copy of a selectable marker is required for viable cell growth, substantial variation in plasmid copy number can go undetected by the MCM assay.

A second type of assay traditionally used to measure plasmid stability uses molecular methods, such as quantitative PCR (qPCR) or Southern blotting, to assess mean plasmid copy number, relative to genomic DNA, across a population of cells (*Lee et al., 2006*). Compared to the MCM assay, qPCR has the advantages of being high-throughput and not requiring a selectable marker in the plasmid of interest. However, qPCR (or Southern blotting) can only measure the average copy

number of a plasmid in a population. Any heterogeneity in plasmid presence or copy number would be undetectable by qPCR. Even a combination of the MCM and qPCR assays lacks the resolution to determine the distribution or variability of plasmid copy number within a host population.

We therefore designed a single-cell assay using a reporter 2μ plasmid. To ensure that this plasmid closely resembles endogenous plasmids, we eschewed the use of the yEP multi-copy plasmids commonly used to express yeast ORFs, because they contain only a small portion of the natural 2μ plasmid. Instead, we built a new 2μ reporter plasmid, which contains both a selectable marker (G418 resistance) as well as a screenable (eGFP) marker, each under a constitutive promoter (*Breslow et al., 2008*). Previously, others described recombinant 2μ plasmid construction and identified a site that does not impact 2μ plasmid stability when less than 3.9 kb DNA is integrated (*Ludwig and Bruschi, 1991*). We therefore integrated the marker cassette (2703 bp) into the endogenous plasmid at this location using yeast assembly (*Figure 1A*; *Ludwig and Bruschi, 1991*; *Gibson et al., 2008*). Importantly, we chose this insertion location because it should not impact typical plasmid function: replication, segregation and copy number amplification should proceed as with the unaltered endogenous plasmid (*Ludwig and Bruschi, 1991*). This allows us to monitor natural plasmid functions relative to variable host compatibility.

These dual markers ensured the reporter plasmid could be both introduced and retained in plasmid-lacking strains. In addition the GFP reporter allows strains to be easily assayed for plasmid presence, absence, and copy number (because GFP intensity scales with copy number in yeast [*Suzuki et al., 2012*; *Labunskyy et al., 2014*; *Lauer et al., 2018*; *Zhu et al., 2015*]). Coupling this reporter with flow cytometry allows us to assay single cells to better understand the dynamics and mechanism of plasmid loss. Our analyses revealed that GFP intensity for this 2μ reporter plasmid is roughly normally distributed across single cells (*Figure 1—figure supplement 1*) indicating that GFP signal did not saturate the detector at high copy number. The endogenous 2μ plasmid loss rate is $\sim10^{-5}$ per cell per generation as estimated by colony-hybridization Southern blots (*Futcher and Cox, 1983*). Based on this prior estimate which relied on total loss events, we infer that the stability of the GFP-2μ reporter plasmid is lower than that of the endogenous 2μ plasmid. This could either reflect the difference in precision of plasmid stability measurements or be due to the cost of constitutive expression of the dual markers. Nevertheless, we conclude that the reporter is well suited for comparative stability studies using the same plasmid in different host backgrounds.

We used this 2μ reporter plasmid with flow cytometry analyses (*Figure 1B*) to simultaneously infer both total plasmid loss events by measuring the proportion of GFP-negative cells, as well as changes in the median plasmid copy number based on GFP intensity (*Figure 1C–D*). Importantly, we could also assess the population distribution of GFP intensity, revealing the inherent cellular heterogeneity in plasmid copy number and loss. This assay is also higher throughput than traditional methods. We refer to this assay as SCAMPR (Single-Cell Assay for Measuring Plasmid Retention).

## 2μ plasmid instability in natural yeast isolates is rare and heritable

2μ plasmids are nearly universally present in laboratory strains of *S. cerevisiae*. However, recent studies of natural isolates have revealed a diversity of plasmid types in natural populations, and even strains lacking 2μ plasmids altogether (*Peter et al., 2018*; *Strope et al., 2015*). We were particularly interested in plasmid-free strains as these might harbor genetic variants that actively inhibit plasmid stability. To this end, we surveyed a panel of 52 natural *S. cerevisiae* isolates for plasmid presence versus absence via PCR analyses (see Materials and methods). From this panel, we identified three strains (representative gel in *Figure 2—figure supplement 1A*) that do not contain the 2μ plasmid: Y9 (from ragi, millet), YPS1009 (from oak exudate), and Y12 (from palm wine) (*Supplementary file 1*). To rule out the possibility that PCR surveys were confounded by 2μ polymorphisms, we also tested these strains via Southern blotting (*Figure 2—figure supplement 1B*), which supported our conclusion of plasmid absence. Wild diploid strains are homothallic, and capable of mating type switching and self-diploidizing following sporulation. To create stable haploid lines for subsequent analyses, we deleted *HO* endonuclease in the natural isolates before sporulating to produce stable heterothallic haploid strains from each of the three plasmid-free natural yeast isolates (*Supplementary file 2*).

Although these three strains lack detectable 2μ plasmids, this absence could be either the result of stochastic loss or host genetic variation that inhibits plasmid stability. Stochastic loss could occur because of rare bottlenecking events in wild populations or during laboratory passaging

(*Kelly et al., 2012*; *Nakayashiki et al., 2005*). We predict that such losses would not protect these strains from re-introduction of natural 2μ plasmids via sex and subsequent propagation (*Futcher and Cox, 1983*). Therefore, if absence were due to stochastic loss, we would expect our reporter 2μ plasmids to be stable in these strains. Alternatively, if the absence of 2μ plasmids reflects true host genetic variation conferring resistance, our reporter 2μ plasmid would be mitotically unstable in these strains. To test these two alternatives, we transformed the GFP-2μ reporter plasmid into haploid cells from these three natural isolates and tested for mitotic plasmid loss using a qualitative colony sectoring assay. As a control, we examined reporter stability in the permissive lab strain BY4742 that was 'cured' of its endogenous 2μ plasmid (*Tsalik and Gartenberg, 1998*) (see

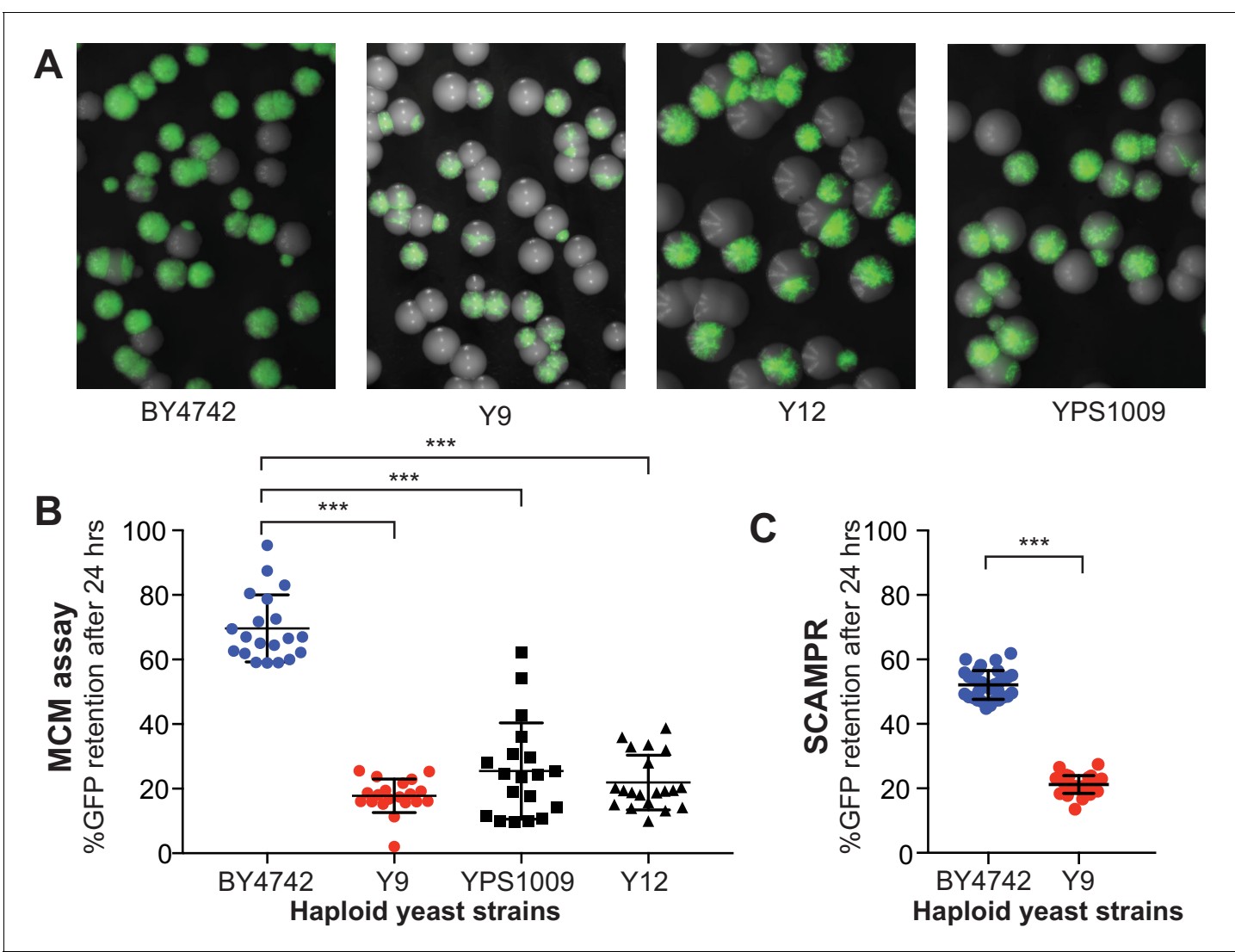

**Figure 2.** Plasmid instability is a heritable trait in three natural *S. cerevisiae* isolates. (A) A colony sectoring assay qualitatively measures GFP-2μ reporter plasmid loss on solid media. Whereas the majority of colonies in the BY4742 background express GFP, only a small fraction of cells in colonies from wild isolates Y9, Y12, and YPS1009 express GFP. (B) The MCM assay quantifies the frequency of 2μ loss events in different yeast strains. Haploid cells from three wild isolates (Y9, Y12, YPS1009) have significantly lower plasmid retention than haploid cells from the laboratory BY4742 strain. ***p<0.0001, Kruskal-Wallis test. (C) SCAMPR assays confirm that a significantly smaller fraction of Y9 strain haploid cells retain the GFP-2μ reporter plasmid after 24 hr, relative to haploid BY4742 cells. ***p<0.0001, Kruskal-Wallis test.

The online version of this article includes the following figure supplement(s) for figure 2:

**Figure supplement 1.** Three natural *S. cerevisiae* isolates lack endogenous 2μ plasmids.
**Figure supplement 2.** BY4742 and Y9 show similar growth rates.

Materials and methods). These analyses showed a clear difference in GFP sectoring (plasmid loss) between the BY4742 laboratory strain and the three natural isolates (*Figure 2A*).

To quantify this difference in plasmid stability between the permissive lab strain and the non-permissive natural isolates, we next measured plasmid stability of the GFP-2μ plasmid over a 24 hr period (~12 generations) using a traditional MCM assay (*Figure 2B*; *Maine et al., 1984*). Consistent with the colony sectoring assay, we determined that the reporter GFP-2μ plasmid is significantly less stable in the naturally $cir^0$ wild isolates than in the plasmid-permissive laboratory strain. For example, the Y9 strain maintained 2μ plasmids in only ~5% of cells on average, whereas the BY4742 lab strain maintained plasmids in ~60% of cells. Even after normalization for phenotypic lag (see Materials and methods), we concluded that Y9 and BY4742 strains retain plasmids at 20% versus 70% frequency, respectively. The other two wild strains showed similar plasmid loss frequencies, with the YPS1009 strain exhibiting more variability between replicates than the other two strains. Taken together, these data suggest that 2μ plasmids are mitotically unstable in these three natural isolates. Thus, the absence of endogenous 2μ plasmids in these strains is the result of host genetic variation rather than stochastic plasmid loss.

## Dominant 2μ plasmid instability in the Y9 strain

Of the three natural isolates in which we observed 2μ plasmid instability, the Y9 strain isolate had the least variable plasmid loss phenotype. Furthermore, in a broad analysis of yeast strains, the Y9 strain was found to be phylogenetically close to the Y12 strain (*Hyma and Fay, 2013*; *Liti et al., 2009*). Based on this phylogenetic proximity, we hypothesized that Y9 and Y12 strains may share the same genetic basis for host-encoded plasmid instability, which might make this genetic determinant easier to identify. We therefore decided to focus on further understanding the phenotypic and genetic basis of plasmid instability in the Y9 strain.

We tested whether growth disadvantages could explain the plasmid instability observed in Y9. This possibility was suggested by our colony sectoring assays (*Figure 2A*) in which some GFP-positive colonies were smaller than GFP-negative colonies. We therefore compared growth rates of BY4742 and Y9, each with and without the 2μ reporter plasmid (*Figure 2—figure supplement 2*). We determined that BY and Y9 haploid strains have similar growth rates to one another without the reporter plasmid. Both Y9 and BY4742 exhibited a similar decrease in growth rate when grown under G418 selection to retain the reporter plasmid. This growth difference could be due to either the presence of the selective drug G418, or due to the fitness cost imposed by the reporter plasmid itself. However, in either condition, Y9 and BY4742 showed similar growth rates to one another. We therefore conclude that plasmid carriage cost is not the predominant cause of the different plasmid instability seen in BY4742 and Y9 strains.

We characterized the putative mechanism of 2μ plasmid instability in the Y9 strain using the SCAMPR assay (*Figure 1*, *Figure 2C*, *Figure 1—figure supplement 1B–C*). We measured the change in distribution of GFP intensity (inferring plasmid copy number changes) among single cells, and total loss events (determined by increase in GFP-negative cells). If the 2μ plasmid were undergoing systematic under-replication due to defects in replication, we might expect an overall and homogenous decrease in median plasmid copy number across the population (*Figure 1D*). Alternatively, if the 2μ plasmid were being missegregated, we might instead see increasing population heterogeneity, with some cells inheriting no plasmid, and their sister cells inheriting twice the number of plasmids as the original mother cell (*Figure 1C*). Others have shown that when cells experience over-amplification of 2μ plasmids those cells stop dividing due to the massive fitness cost, as in the case of *nibbled* and similar phenotypes (*Dobson et al., 2005*; *Zhao et al., 2004*; *Zhao and Blobel, 2005*). This fitness defect explains why we may see an increase in plasmid-negative cells at the population level, without the corresponding increase of super-green high-plasmid cells (dotted line).

In our SCAMPR analyses, we find that even under G418 selection, Y9 cells do not maintain the reporter 2μ plasmid as efficiently as BY strains (52% GFP-positive versus 90% respectively) (*Figure 1—figure supplement 1*). Moreover, upon removing pressure to maintain the plasmid (no G418 selection), the proportion of Y9 cells with no GFP (no 2μ plasmid) increases significantly, from 48% to 83%. However, the median GFP intensity (and inferred 2μ plasmid copy number) of plasmid-bearing Y9 cells remains largely unchanged (*Figure 1—figure supplement 1B*); even with G418 selection, GFP intensity (2μ plasmid copy number) is lower in Y9 than BY4742. We therefore conclude that 2μ plasmid loss in Y9 haploid cells occurs primarily via abrupt, complete loss of plasmids from

cells in the population rather than a steady decrease in copy number (*Figure 1—figure supplement 1B*). This observed pattern of plasmid loss is consistent with plasmid segregation failure during mitosis, rather than a copy number suppression mechanism or plasmid under-replication. As a result of this segregation failure, 'non-permissive' Y9 haploid cells lose the 2μ reporter plasmid substantially more quickly than the permissive BY4742 laboratory strain (*Figure 2C*), mirroring our observations from colony sectoring assays (*Figure 2A*).

Next, we investigated whether mitotic instability of the 2μ plasmids in the Y9 strain is genetically recessive or dominant by examining heterozygous diploids of permissive and non-permissive strains. To ensure that mitotic instability was not influenced by ploidy itself, we first tested whether the plasmid instability phenotype we observed in haploid strains persists in homozygous diploid BY4742 and Y9 strains. We found an even bigger difference in plasmid instability between homozygous diploid Y9 and BY4743 strains than between haploid strains (*Figures 3A* and *2B*). We generated a heterozygous diploid strain by crossing the GFP-2μ plasmid-containing permissive BY4742 lab strain to the non-permissive Y9 haploid strain. If plasmid loss in Y9 cells were due to inactivating mutations within a host 'permissivity' factor required for 2μ mitotic segregation, we might expect plasmid instability to be recessive, with the BY4742 allele providing rescue in the heterozygote. Alternatively, if plasmid instability in Y9 cells were due to a host-encoded, gain-of-function 'restriction' factor that impairs mitotic stability of 2μ plasmids, we would expect mitotic instability of 2μ plasmids to be dominant; heterozygous diploids would also exhibit plasmid instability. We found that heterozygous BY4742/Y9 diploid cells rapidly lose the plasmid after G418 selection is removed (*Figure 3A*). These findings could result from haploinsufficiency of a permissivity factor, or dominance of plasmid restriction factors in the Y9 genome. We therefore considered both possibilities in subsequent analyses.

## Genetic architecture underlying 2μ plasmid instability in the Y9 strain

Previous studies have shown that 2μ plasmids efficiently propagate via non-Mendelian inheritance through meiosis in laboratory strains of *S. cerevisiae* (*Harrison et al., 2014*; *Sau et al., 2014*; *Brewer and Fangman, 1980*; *Hsiao and Carbon, 1981*). Because BY4742/Y9 heterozygous diploids exhibit dominant plasmid loss, we maintained G418 selection up to and during sporulation to enrich for tetrads in which all four spores retained 2μ reporter plasmids. We then measured plasmid instability phenotypes among meiotic progeny of BY4742/Y9 heterozygous diploids to understand the genetic architecture underlying the Y9 strain's plasmid instability.

If a single genetic locus were responsible for 2μ plasmid instability, we would expect tetrads to exhibit a 2:2 segregation pattern, with half of the spores phenotypically resembling the permissive BY4742 parent and the other half resembling the non-permissive Y9 parent. Of the 60 tetrads examined, approximately 20% of 4-spore tetrads exhibited a roughly 2:2 segregation pattern, and the remaining 80% tetrads exhibited more complex patterns of inheritance (*Figure 3—figure supplement 1*). Our results indicate that plasmid instability is heritable but not monogenic. Based on these findings, we used the Castle-Wright estimator (*Lande, 1981*; *Zeyl, 2005*) to estimate that 2μ plasmid instability is encoded by at least 2–3 independently segregating large effect loci in the Y9 genome, although these estimates could be affected by stochastic variability of the 2μ plasmid stability phenotype.

Next, we performed quantitative trait locus (QTL) mapping using bulk segregant analysis (BSA) to identify genetic loci that contribute to the Y9 strain's 2μ plasmid instability phenotype (*Figure 4—figure supplement 1*; *Ehrenreich and Magwene, 2017*; *Lander and Botstein, 1989*). We selected 600 random spores resulting from a heterozygous BY4742/Y9 diploid containing our reporter 2μ plasmid and used SCAMPR to phenotype plasmid stability (*Amberg et al., 2005*). Most of these progeny exhibit intermediate 2μ plasmid stability between haploids of the parental BY4742 and Y9 strains (*Figure 3B–C*). We then pooled and bulk-sequenced 132 'non-permissive' progeny strains that represented ~20% of progeny with the lowest 2μ plasmid stability and 126 'permissive' strains that represented the ~20% of progeny with the highest 2μ plasmid stability (*Figure 3C*).

In addition to the progeny pools, we also sequenced the genomes of the three 2μ plasmid-negative strains we identified (Y9, Y12, and YPS1009), as well as UC5, a 2μ plasmid-containing strain (by PCR) that is closely related to Y9 and Y12 (*Hyma and Fay, 2013*; *Cromie et al., 2013*). We mapped reads from these strains back to the *S. cerevisiae* reference genome and created de novo assemblies for each strain (see Materials and methods). Unexpectedly, whole genome sequencing revealed that the haploid Y9 parent strain was disomic for chromosome XIV (*Figure 4—figure supplement 2A*),

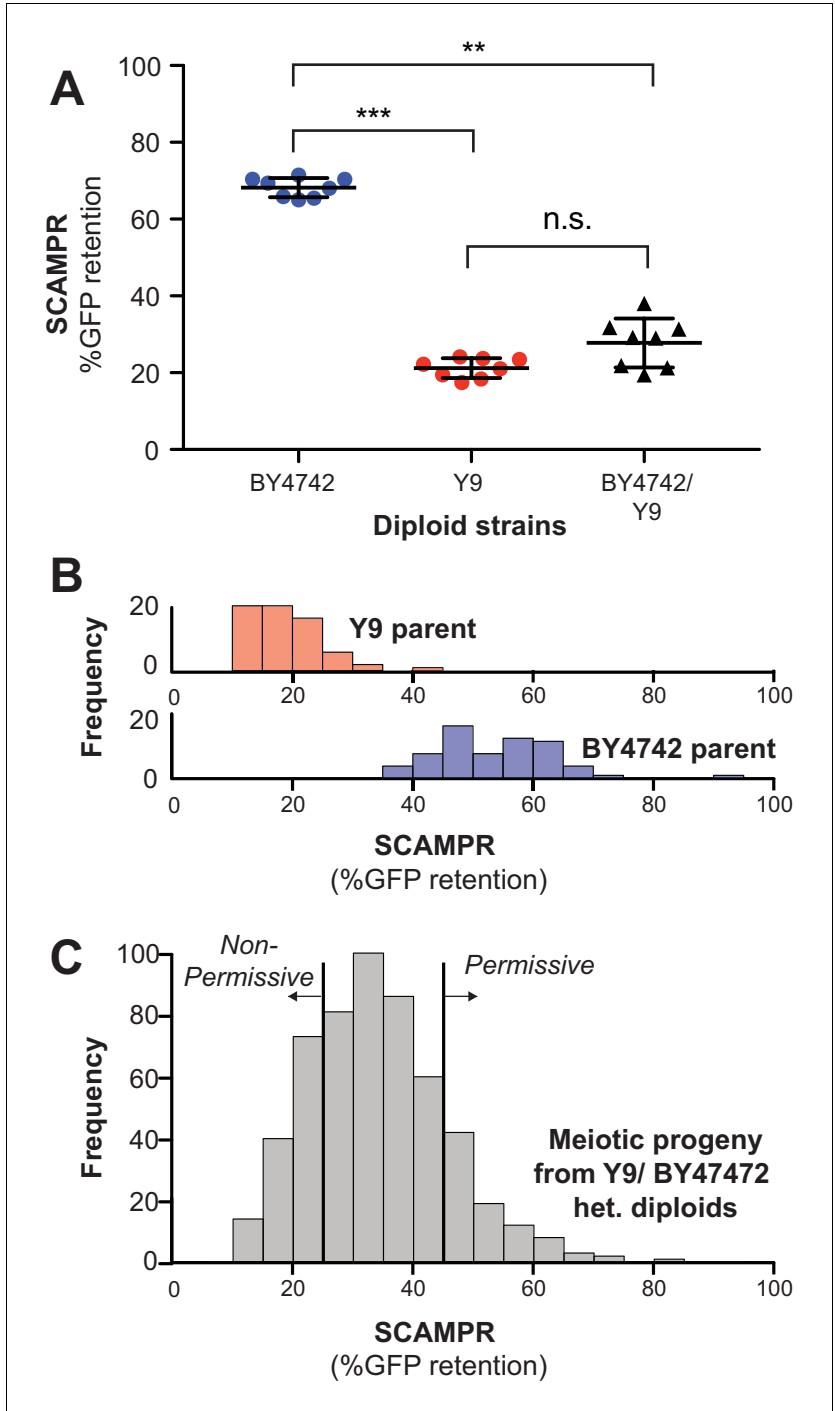

**Figure 3.** Genetic architecture and dominance of the Y9 plasmid instability phenotype. (**A**) Compared to homozygous BY4742 diploids, heterozygous BY4742/Y9 diploid cells display low plasmid retention after 24 hr, similar to homozygous Y9 diploids. This suggests that the plasmid instability of Y9 cells is a dominant trait. All strains were analyzed with the SCAMPR plasmid retention assay. \*\*p<0.001, \*\*\*p<0.0001, Kruskal-Wallis test; n. s. = not significant. (**B–C**) Phenotype distribution across ~600 random progeny strains (**C**) shows that most have an intermediate phenotype between that of the parental haploids (**B**). All strains were analyzed in triplicate with the SCAMPR assay. We selected the bottom ~20% ('non-permissive') and top ~20% ('permissive') of strains from this distribution for bulk sequencing and segregant analysis.

The online version of this article includes the following figure supplement(s) for figure 3:

*Figure 3 continued on next page*

*Figure 3 continued*

**Figure supplement 1.** Tetrads dissected from meiosis of B4742/Y9 heterozygous diploids reveal a range of plasmid stability phenotypes.

with the aneuploid chromosome segregating in the Y9 x BY4742 cross. The homothallic Y9 diploid was euploid for chromosome XIV by qPCR and shows a similar plasmid loss phenotype as a homozygous diploid Y9 that has an additional chromosome XIV (*Figure 4—figure supplement 2B*). These data demonstrate that the aneuploidy for chromosome XIV is not a large contributor to Y9's plasmid instability phenotype (*Pavelka et al., 2010*). We therefore disregarded the segregating chromosome XIV disomy in our subsequent analyses.

We identified genomic differences between the Y9 and BY4742 strains (see Materials and methods), then compared allele frequencies between the 'permissive' and 'non-permissive' pools of meiotic progeny from BY4742/Y9 heterozygotes (*Figure 3C*, *Figure 4—figure supplement 1A*). We identified genomic regions in which inheritance of the Y9 allele is significantly more common in the non-permissive progeny pool than the permissive pool (*Figure 4A*) using the MULTIPOOL algorithm to generate likelihood-based 'LOD' (logarithm of the odds) scores (*Lander and Botstein, 1989*; *Edwards and Gifford, 2012*). We identified loci that are likely linked to the plasmid stability phenotype. Although there are a few genomic regions with moderate LOD scores of ~4 (*Figure 4B*), the most striking LOD score of 9.996 was seen for a high-confidence QTL on chromosome V. While it is challenging to establish a concrete LOD score threshold above which loci are statistically significant, a score of ~10 is comfortably above genome-wide significance thresholds of 3.1–6.3 established empirically in other studies (*Treusch et al., 2015*; *Albert et al., 2014*; *Roberts et al., 2017*). This locus likely encodes the strongest genetic determinant of plasmid instability in the Y9 genome.

## A single variant of the essential SUMO ligase *MMS21* contributes to 2μ mitotic instability in Y9

We focused our efforts on variants within the QTL on chromosome V to identify the genetic basis of Y9-encoded plasmid instability (*Figure 4—figure supplement 3*). The 90% confidence interval for this QTL is ~91 kb wide and contains 54 ORFs, with a 50% confidence interval 23 kb wide (16 ORFs) (*Figure 4—figure supplement 3*, *Figure 4C*). This region contains many polymorphisms between the Y9 and BY4742 genomes but very few structural variants (*i.e.,* large insertions, deletions, translocations). One of these structural variants is the *URA3* gene, which is present in Y9 and was specifically deleted in BY4742. Although the *URA3* gene often falls within fitness-related QTLs in BY4742 crosses, detailed follow-up studies (*Figure 4—figure supplement 4*) allowed us to conclusively rule out a role for *URA3* in the plasmid instability phenotype of the Y9 strain (*Wilkening et al., 2014*; *Romano et al., 2010*).

After excluding dubious ORFs, 44 *bona fide* protein-coding genes remained within the 90% confidence interval. 28 of these candidate genes contained a total of 94 missense changes between Y9 and BY4742, while the rest contained no non-synonymous differences between the parental strains for our QTL cross. We focused on 15 missense polymorphisms (in 11 genes) at which Y9 is identical to the phylogenetically close non-permissive strain, Y12, but different from the closely-related permissive strain, UC5 (*Supplementary file 3*). Our attention was drawn to *MMS21*, which contains a single Thr69Ile missense change common to Y9 and Y12, but distinct from the BY4742 laboratory strain and the permissive UC5 strain. This polymorphism is only 1.2 kb away from the apex of the LOD score peak in our bulk segregant analysis. Even though *MMS21* has not previously been implicated in 2μ biology, it encodes one of the three mitotic SUMO E3 ligases in *S. cerevisiae* (*Zhao and Blobel, 2005*). The two other SUMO E3-ligases, encoded by *SIZ1* and *SIZ2,* have known roles in SUMO-modification of the plasmid-encoded Rep and Flp1 proteins to cause instability or hyper-amplification phenotypes (*Dobson et al., 2005*; *Pinder et al., 2013*; *Chen et al., 2005*). Therefore, we evaluated the consequences of Y9's *MMS21* polymorphism on 2μ plasmid mitotic stability.

We first tested whether the Y9 *MMS21* allele was sufficient to confer the plasmid loss trait. We integrated the Y9 *MMS21* allele, with the flanking Y9 intergenic regulatory regions, into the *ho* locus of BY4742. These engineered BY4742 haploids thus express both the BY4742 and Y9 *MMS21* alleles.

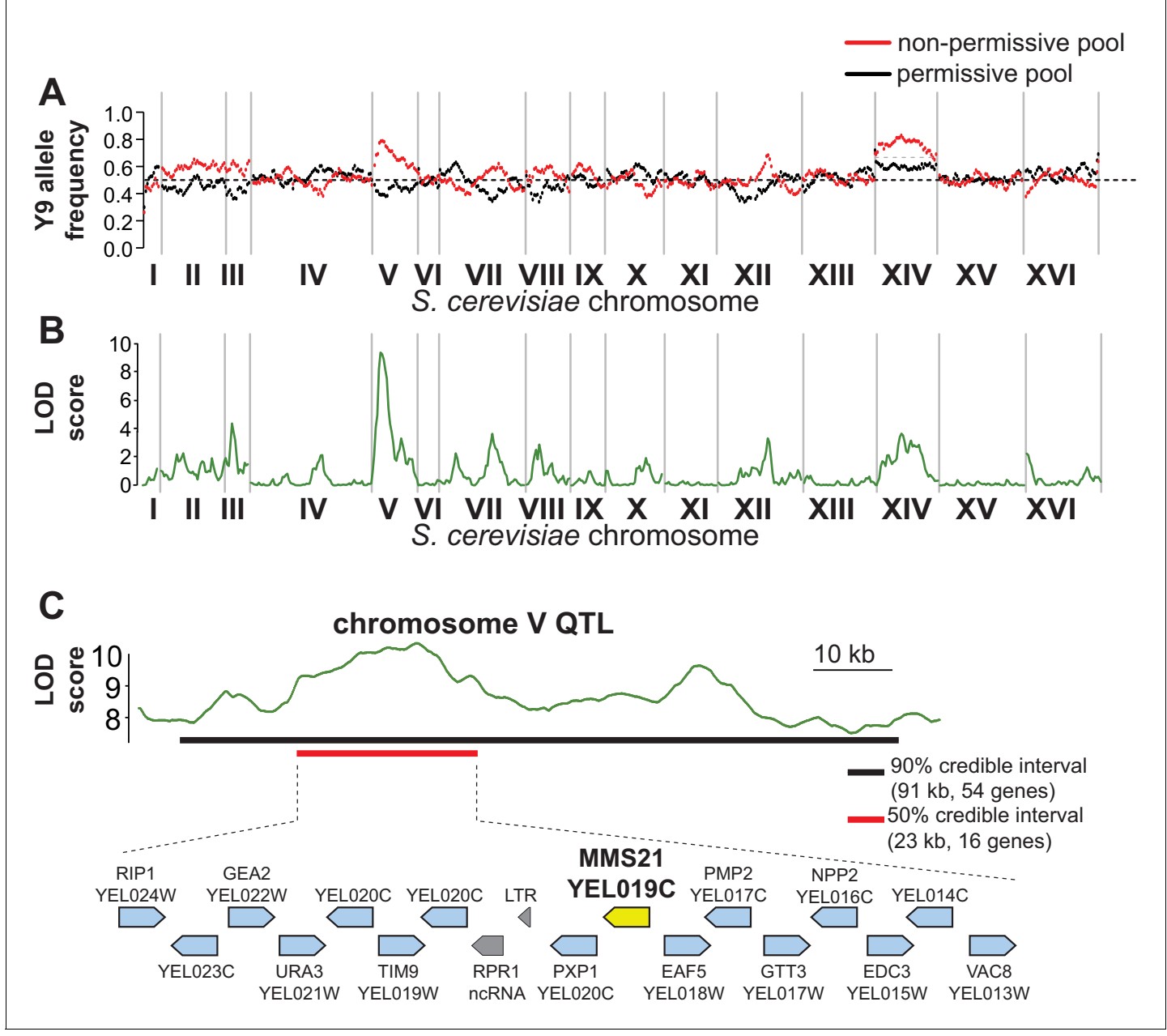

**Figure 4.** QTL mapping identifies a plasmid instability locus on Y9 chromosome V. (**A**) We plotted the mean Y9 SNP allele frequency in 20 kb windows for the 'non-permissive' (red) and 'permissive' (black) pools of meiotic haploid progeny from BY4742/Y9 heterozygous diploid parents. Associations with a plasmid instability locus would show an increased representation of Y9 alleles in the non-permissive pool and a decreased representation of the BY4742 haplotype in the permissive pool (dotted line indicates equal representation). The increased representation of Y9 alleles on chromosome XIV in both pools is a result of a segregating disomy in the Y9 parent that we show does not affect the plasmid instability phenotype (*Figure 4—figure supplement 2*). (**B**) Based on the allele frequencies of individual SNPs, we used MULTIPOOL to calculate LOD scores for association with the plasmid instability phenotype. We observe a highly significant LOD score (10.00) on chromosome V. The peak is fairly sharp and reaches maximal LOD score at chrV:122.3–122.7 kb (sacCer3 coordinates). All loci have allele frequencies skewed in the expected direction; the restrictive pool is enriched for Y9 alleles. (**C**) MULTIPOOL 90% (54 genes, 91.2 kb, chrV:92.4–183.6 kb) and 50% (16 genes, 23.1 kb, chrV:107.3–130.4 kb) credible intervals for the chromosome V QTL. Among the 16 genes in the 50% credible interval is *MMS21*, which encodes an essential SUMO E3-ligase.

The online version of this article includes the following figure supplement(s) for figure 4:

**Figure supplement 1.** Schematic of QTL mapping by bulk segregant analysis (*Magwene et al., 2011*).

**Figure supplement 2.** Aneuploidy of chromosome XIV in Y9 strain.

**Figure supplement 3.** Y9 chromosome V is most strongly associated with plasmid instability.

**Figure supplement 4.** Deletion of URA3 from Y9 haploid cells does not affect their plasmid instability phenotype.

Although the addition of Y9 *MMS21* does lower plasmid stability, this difference is not statistically significant from BY4742 haploids that only express the BY4742 allele (*Figure 5A*). Thus, the Y9 *MMS21* allele, by itself, does not appear to be sufficient to lower plasmid stability in the BY4742 genetic background.

Next we used reciprocal hemizygosity to test whether loss of Y9 *MMS21* from heterozygous BY4742/Y9 diploids would lead to an increase in plasmid stability. Because *MMS21* is an essential gene, we could not simultaneously delete both the Y9 and BY4742 *MMS21* alleles in heterozygous diploids. Instead, we deleted either the Y9 or the BY4742 allele, yielding BY4742/Y9 diploids that are hemizygous for one or the other *MMS21* allele. If the plasmid stability phenotype were affected by *MMS21* haploinsufficiency, we would expect that deletion of either *MMS21* allele would affect the stability phenotype. Contrary to this expectation, we found that deletion of the Y9 *MMS21* allele, but not the BY4742 allele, results in a reproducible and statistically significant increase in plasmid stability of nearly 8% (*Figure 5B*). Indeed, otherwise identical heterozygous Y9/BY4742 strains that are hemizygous for either the Y9 or the BY4742 allele of *MMS21* differ significantly in their plasmid instability phenotype (*Figure 5B*). Our results show that *MMS21* allelic differences contribute significantly to the 2μ instability of the Y9 strain. These data are also consistent with Y9 plasmid instability being due to dominant plasmid restriction, rather than a haploinsufficient permissivity factor. However, *MMS21* does not explain the entire Y9 phenotype, consistent with plasmid instability segregating as a multigenic trait through the cross. The remaining trait-determining loci in Y9 likely include some of the minor QTL peaks we found but could also include linked polymorphisms within the chromosome V region.

The phenotypic difference between hemizygous strains (*Figure 5B*) could be due to the Thr69Ile coding polymorphism found in Y9 and Y12 strains relative to BY4742, or due to regulatory differences, or both. We applied the same comparative genomics approach that initially identified Thr69Ile

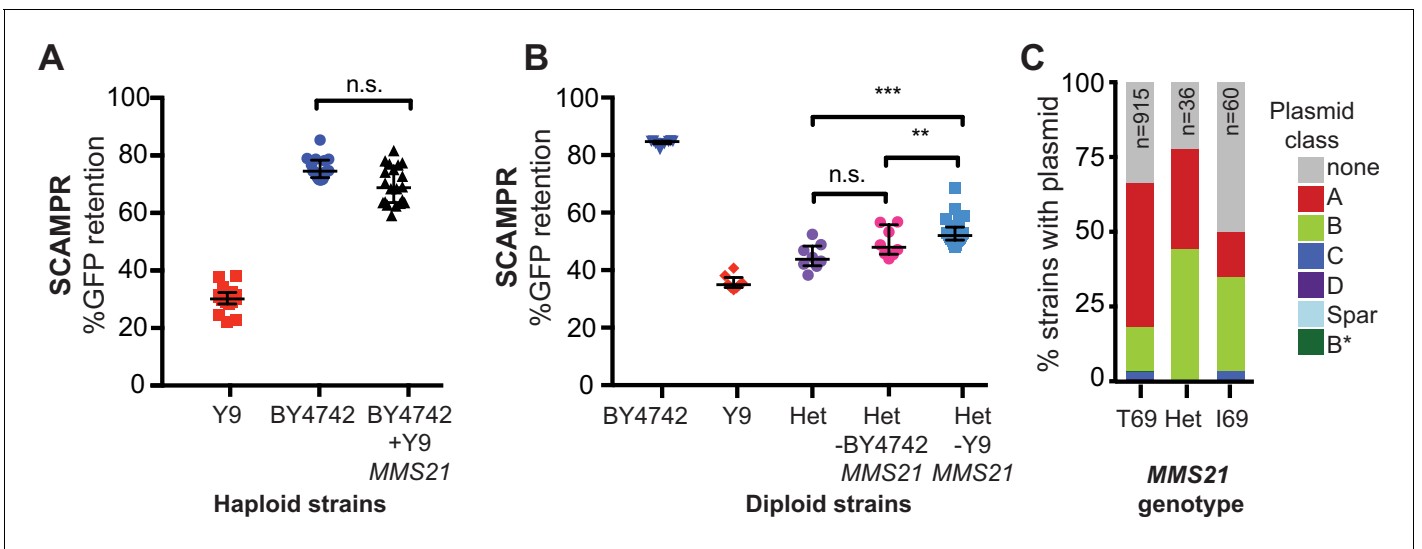

**Figure 5.** A single SNP in Y9 *MMS21* contributes to the plasmid instability phenotype. (A) Introduction of the Y9 *MMS21* allele into BY4742 haploid cells is not sufficient to significantly lower plasmid instability. (B) However, removal of the Y9 *MMS21* allele but not the BY4742 *MMS21* allele increases plasmid stability in BY4742/Y9 heterozygous diploids, showing that the Y9 allele of *MMS21* plays an important role in the Y9 plasmid instability phenotype. **p<0.001, Kruskal-Wallis test, n.s. = not significant. (C) Plasmid prevalence (by plasmid class) for each *MMS21* Thr9Ile genotype within 1011 sequenced *S. cerevisiae* strains. Plasmid data and genotypes from *Peter et al., 2018*. Strains with the Y9 *MMS21* allele (I69) have a lower frequency of harboring 2μ plasmids in general, and A-type 2μ plasmids, in particular. However, this effect can be confounded by the phylogenetic relatedness of these strains.

The online version of this article includes the following figure supplement(s) for figure 5:

**Figure supplement 1.** Comparative analysis of MMS21 and flanking regions.

**Figure supplement 2.** Sequence of *MMS21* codon 69 across the *Saccharomyces* sensu stricto clade, as well as selected *S. cerevisiae* strains and two outgroup *Naumovozyma* species.

**Figure supplement 3.** The Thr69Ile polymorphism is located at the Mms21-Smc5 binding interface.

to the intergenic (candidate regulatory) regions flanking *MMS21*. In addition to the missense polymorphism, the Y9 strain differs from BY4742 at four synonymous sites within the *MMS21* ORF and at a total of 11 sites in the two flanking regions (*Figure 5—figure supplement 1*). Y9 and Y12 are identical at all 16 of these sites. We next examined the closest outgroup strain, UC5, which still bears plasmids according to our PCR survey (*Figure 2—figure supplement 1B*). We found that UC5 differs from Y9 and Y12 at only three sites in the *MMS21* locus: two synonymous SNPs and the single missense SNP. Thus, only these three sites strictly correlate with the plasmid instability phenotype, whereas the intergenic SNPs do not. Based on this finding, we chose to focus on the *MMS21* Thr69Ile variant. However, regulatory differences between Y9 and BY4742 may still contribute to natural variation in plasmid stability.

### *MMS21* natural variation within *S. cerevisiae* and between sensu stricto species

The Thr69Ile change found in Y9 and Y12 strains is not found in the third non-permissive strain, oak YPS1009, suggesting that YPS1009 acquired 2μ plasmid instability through an independent evolutionary path. The Thr69 allele found in the BY4742 lab strain appears to be the ancestral allele, with Ile69 arising more recently in a subset (96) of 1,011 *s. cerevisiae* strains that were sequenced as part of a recent large-scale study (*Peter et al., 2018*). This study also reported which of the 1011 strains carry 2μ plasmids. Upon reanalyzing these data, we find that a smaller proportion of *S. cerevisiae* strains homozygous for the *MMS21* Ile69 allele harbor 2μ plasmids compared to strains homozygous for the ancestral Thr69 allele (*Figure 5C*). In particular, the A-type 2μ plasmids, which we have tested using SCAMPR in this study, appear to be particularly depleted in strains with the Ile69 allele, suggesting that this allele might specifically restrict A-type 2μ plasmids (*Figure 5C*). While interesting, at present we cannot distinguish whether these observations are a result of a causal association or of shared evolutionary history, due to phylogenetic relatedness of the Ile69 allele-encoding strains.

To explore natural variation in *MMS21* beyond *S. cerevisiae*, we aligned sequences from selected *S. cerevisiae* strains as well as other *Saccharomyces* sensu stricto species and two outgroups (*N. castelli* and *N. dairenensis*) (*Figure 5—figure supplement 2*). Interestingly, *S. eubayanus* and *S. uvarum* also seem to have independently acquired Ile69 but still harbor endogenous 2μ plasmids, whereas *S. arboricola* has yet another amino acid (alanine) at this position (*Strope et al., 2015*).

The location of the Y9/Y12 amino acid change in the Mms21 protein also provides important clues to its functional consequences. The Thr69Ile change occurs in the third of three alpha-helices in the Mms21 N-terminal domain, which makes contact with the Smc5/6 complex, and is essential for yeast viability (*Duan et al., 2009*; *Figure 5—figure supplement 3*). Yeast cells deficient for *MMS21* show gross chromosomal segregation defects and die as large, multi-budded cells (*Bermúdez-López et al., 2010*). However, the C-terminal zinc finger RING domain responsible for sumoylation of substrates is dispensable for Mms21's essential function (*Duan et al., 2009*). We therefore speculate that the non-permissive *MMS21* allele may act by directly affecting the Smc5/6 complex rather than through its sumoylation function. However, these possibilities may be hard to distinguish because the SUMO ligase function of Mms21 also depends on its docking with the Smc5/6 complex (*Bermúdez-López et al., 2015*). We also examined the Y9 strain for polymorphisms in other members of the Smc5/6 complex. While there are other polymorphisms in Smc5/6 complex members, we observed none in the regions of Smc5 that interact with any segment of Mms21 (*Duan et al., 2009*; *Supplementary file 4*). Despite the Smc5/6 complex's essential role in the removal of DNA-mediated linkages to prevent chromosome missegregation and aneuploidy, it has not been directly implicated in 2μ stability. Our finding that a single polymorphism at the Mms21-Smc5 interaction interface reduces 2μ stability thus reveals a novel facet of host control.

## Discussion

In this study, we leveraged natural variation to identify a gain-of-function variant that restricts 2μ plasmids in *S. cerevisiae*. Our approach is complementary to the traditional biochemical and genetic approaches that have previously used loss-of-function genetic analyses to study host regulation of 2μ plasmids. Natural variation studies can identify alleles of host genes that retain host function but still block SGEs like 2μ plasmids. Such studies can reveal novel mechanisms of host control, which may be otherwise challenging to discover via loss-of-function analysis.

Although 2μ-based vectors have long been used as an important tool in yeast genetics, studies of 2μ plasmids as natural SGEs have lagged behind considerably. Our new phenotyping assay, SCAMPR, makes the 2μ plasmid a more tractable system. SCAMPR captures single-cell data that facilitate studies of population heterogeneity, allowing inferences of the mechanisms by which plasmids may be controlled by their hosts. Although recent advances in single-cell genome sequencing make it possible to directly sequence and infer copy number of 2μ plasmids, this would be prohibitively expensive compared to the GFP-based flow cytometry profiling methodology we use. SCAMPR has potential for expanded use, for example to explore meiotic plasmid transmission dynamics. SCAMPR could also be paired with host lineage tracking to assess plasmid fitness burden alongside plasmid loss dynamics in competitive fitness assays. This paired strategy would provide a powerful approach for understanding the relative contribution of both plasmid fitness cost and host-plasmid incompatibility across hosts. In general, SCAMPR could be utilized to study high-copy number SGE plasmid dynamics, DNA replication and segregation, in any system where expression is well matched to copy number.

Our survey of 52 wild *S. cerevisiae* isolates identified three strains that naturally lack 2μ plasmids. Detailed studies of one of these strains, Y9, revealed that 2μ plasmid instability is heritable, dominant and likely the result of multiple contributing alleles. Through QTL mapping by bulk segregant analysis, we identified a significant locus on chromosome V associated with 2μ plasmid loss. We found that a single amino acid variant in Y9 *MMS21*, which encodes an essential SUMO E3 ligase in *S. cerevisiae*, contributes to 2μ plasmid instability. *MMS21* does not fully account for the 2μ plasmid loss phenotype in heterozygous BY4742/Y9 strains. This result is unsurprising based on our tetrad analysis and QTL mapping, which both suggest that additional independently segregating loci affect plasmid stability. Although loss of Y9 *MMS21* from heterozygous diploids leads to a relatively modest effect on 2μ plasmid instability, it may still account for all of the QTL signal we observe in chromosome V. Alternatively, the QTL on chromosome V could contain additional determinants of plasmid instability in close genetic linkage to *MMS21*, either in coding or regulatory sequences. CRISPR-Cas9 based approaches will be useful to test a large number of genomic changes rapidly and in parallel between Y9 and BY4742 to identify other determinants of plasmid instability in this QTL and in other candidate loci (*Sadhu et al., 2016*; *Sharon et al., 2018*).

The reciprocal hemizygosity experiment (*Figure 5B*) reveals that the Y9 variant of *MMS21* acts dominantly to restrict mitotic stability of 2μ plasmids, ruling out the possibility that this allele is haploinsufficient. We considered two scenarios by which this allele may exert its dominant effect on plasmid stability. The first scenario is that the Y9 allele of *MMS21* encodes a dominant-negative allele, which impairs the function of the Smc5/6 complex whether in a haploid or heterozygous state. Although this impairment does not negatively impact essential host functions when 2μ plasmid is absent (Y9 strains are viable and fit), this allele would have a fitness deficit in the presence of 2μ plasmids, as host functions become overburdened when hijacked by the parasite. Under this scenario, we would expect to see a greater fitness loss due to 2μ plasmid presence in Y9 compared to BY4742. However, both Y9 and BY4742 strains suffer an equal fitness loss in the presence of 2μ reporter plasmids (*Figure 2—figure supplement 2*). Moreover, plasmid-restrictive host alleles would only arise and propagate in natural populations if their fitness cost did not outweigh the modest 1–3% fitness cost imposed by the widespread 2μ plasmids in *S. cerevisiae* populations. We therefore favor a second scenario, in which the Y9 *MMS21* allele represents a separation-of-function allele that is still capable of performing host functions but impairs 2μ mitotic stability.

The 2μ plasmids appear to have co-evolved with budding yeasts for millions of years and are prevalent in species such as *S. cerevisiae*. Long-term coevolution appears to have 'optimized' 2μ plasmids as tolerable parasites: not too great of a burden on host fitness, but still high enough plasmid copy numbers to ensure stable propagation. This copy number balance is achieved through both plasmid (*e.g.,* Flp1 repression) and host (e.g., sumoylation) contributions. Nevertheless, there are hints that this truce between 2μ plasmids and yeast may be uneasy. 2μ plasmid stability is frequently compromised in heterospecific (other species) hosts, suggesting it is actively adapting to maintain stability within its native host species (*Murray et al., 1988*). Our discovery of a natural host variant of *S. cerevisiae* that impairs conspecific (same species) 2μ plasmid stability further supports the hypothesis that even the low fitness costs imposed by 2μ plasmids are sufficient to select for host evolutionary resistance.

Most studies of 2μ plasmid plasmids (including this one) have focused on the A-type variant that is most commonly found in laboratory strains. However, new sequencing studies have revealed that *S. cerevisiae* strains harbor a diverse set of 2μ plasmid plasmids (*Peter et al., 2018*; *Strope et al., 2015*). This diversity of 2μ plasmid plasmids might itself have arisen as a result of host defenses within *S. cerevisiae*, leading to plasmid diversification. For instance, although SCAMPR studies revealed the importance of the Y9 *MMS21* variant against the stability of the A-type plasmid, it is possible that this variant is ineffective against the other 2μ genotypes. Thus, 2μ plasmids might exist in a frequency-dependent regime with their budding yeast hosts; A-type plasmids might thrive in certain host genetic backgrounds whereas B-type plasmids might thrive in others. The simultaneous presence of multiple 2μ plasmid types within species could explain the presence of standing variation in plasmid instability phenotypes in *S. cerevisiae* populations, including the low observed frequency of the Y9 *MMS21* allele.

Testing the effects of *MMS21* and other restrictive alleles on stability of different 2μ plasmids (e.g. B- or C-type) would provide a means to distinguish between universal versus plasmid-type-specific restriction. Future studies could employ SCAMPR to study the functional consequences of the natural diversity of 2μ plasmids in yeast. In particular, SCAMPR reporters from different types of *S. cerevisiae* 2μ plasmids and from divergent *Saccharomyces* species may reveal important biological determinants behind their co-evolution and long-term success in budding yeast species.

The identification of *MMS21* led us to initially suspect that this locus might represent another connection between the SUMO-ligation machinery and 2μ plasmid stability (*Dobson et al., 2005*; *Zhao et al., 2004*; *Pinder et al., 2013*; *Ma et al., 2019*). However, the location of the Thr69Ile missense change at the binding interface between Mms21 and Smc5 (*Figure 5—figure supplement 3*) suggested a mechanism that relies on the Smc5/6 complex rather than the catalytic RING domain. Interestingly, a recent study demonstrated that even the SUMO ligase function of Mms21 depends on its docking with the Smc5/6 complex (*Bermúdez-López et al., 2015*). Thus, the polymorphism in the Mms21-Smc5 interaction site could either affect Mms21's SUMO ligase function or other functions of the Smc5/6 complex. The Smc5/6 complex lies at the nuclear periphery, where it anchors dsDNA breaks to facilitate repair, resolves X-shaped DNA structures that arise during DNA replication and repair, and helps mediate sister chromatid cohesion (*Bermúdez-López et al., 2010*). All three of these cellular processes might directly impact stability of 2μ plasmids (*Velmurugan et al., 2000*). Alteration of Mms21 function, an essential component of the Smc5/6 complex, could thus directly affect both segregation of 2μ plasmids as well as interfere with their amplification via Flp1-induced recombination intermediates.

Although Smc5/6 has not been previously implicated in 2μ stability, this complex is involved in the stability of viral episomes, as human Smc5/6 acts as a restriction factor against hepadnaviruses such as human Hepatitis B virus (*Decorsière et al., 2016*; *Murphy et al., 2016*). To counteract this restriction function, diverse hepadnaviruses encode antagonist HBx proteins that degrade mammalian Smc5/6 and restore viral fitness (*Decorsière et al., 2016*; *Murphy et al., 2016*; *Abdul et al., 2018*). Our findings that components of the yeast Smc5/6 complex affect 2μ stability suggest that the Smc5/6 complex might provide a general mechanism to protect host genomes from the deleterious consequences of multicopy genetic parasites.

## Materials and methods

### Strain growth and construction

Yeast strains were grown in standard yeast media at 30°C unless otherwise noted (*Amberg et al., 2005*). Transformations were carried out using a high-efficiency lithium acetate method (*Amberg et al., 2005*). The GFP-2μ plasmid was created by Gibson assembly directly into otherwise plasmid-less yeast strains *cir⁰* BY4741 (*MATa* haploid) and BY4742 (*MATα* haploid), which had been cured of their endogenous plasmids by previously published methods (*Tsalik and Gartenberg, 1998*; *Gibson, 2011*). To avoid disruption of the plasmid's endogenous replication and segregation machinery, a cassette containing both markers was integrated into the A-type 2μ sequence found in the *S. cerevisiae* laboratory strain BY4741 at a restriction site reported to tolerate insertions of up to 3.9 kb without impacting copy number or stability (*Ludwig and Bruschi, 1991*). We did not use any bacterial cloning vector sequences to minimize unnecessary or destabilizing changes to the 2μ

reporter plasmid, so the reporter plasmid was directly assembled in yeast. Assembling in $cir^0$ strain backgrounds avoided multiple plasmid genotypes within a strain background that could have led to plasmid competition or recombination.

We used the NEBuilder HiFi DNA Assembly Master Mix (product E2621) for Gibson assemblies. Yeast plasmids were recovered using Zymoresearch Zymoprep Yeast miniprep kits (D2004). The assembled plasmids were then retransformed to the same $cir0$ yeast backgrounds to ensure plasmid clonality. Genetic crosses were carried out on a Singer Sporeplay dissection scope, for both tetrad dissection and selection of unique zygotes for mating strains. Strain mating type was confirmed by halo formation in the presence of known mating type tester strains. Strains used in this work are listed in *Supplementary file 1*.

Natural isolates were obtained as homothallic diploids (capable of mating type switching and self-diploidization). We made stable heterothallic haploid strains (no longer capable of mating type switching) by first knocking out *ho* endonuclease prior to sporulation (*hoΔ::HphNT1*). We found that the natural isolates required significantly longer homology arms for proper DNA targeting when making integrated genomic changes (e.g. gene deletions) via homologous recombination. Where BY4742 lab strains utilized ~50 bp homology arms for high efficiency recombination, Y9 required ~1 kb flanking homology. Even with longer homology, a substantial number of clones in any transformation did not contain the desired edit. These hurdles made editing the Y9 genome challenging.

## Growth rate measurements

Growth rate measurements were performed in 96-well plates in Biotek Powerwave incubating plate readers with Gen5 software at 30°C with shaking. Logarithmically dividing cells were seeded at approximately 2000 cells in 100 ul of defined medium per well, with or without selection as indicated. Synthetic complete media was prepared with monosodium glutamate as the nitrogen source to facilitate G418 selection. Twelve replicate wells were run for each strain background and $OD_{660}$ measurements were taken every 10 min until each well reached saturation. Data were trimmed to time points during log phase growth, and replicate data points are plotted as the mean value of replicates, with error bars indicating SD. The outside wells of the plate were incubated as media blanks to measure baseline $OD_{660}$ and to help monitor and minimize evaporation during the runs.

## Colony sectoring

Confirmed transformants were cultured under G418 selection, then plated to YPD medium where colonies were allowed to form without selective pressure to maintain the reporter plasmid. After 2 days growth at 30°C, colonies were imaged under white light and GFP excitation to assess qualitative plasmid loss in the different strains using a Leica M165 FC dissection scope with a GFP filter and Leica DFC7000 T camera. Colony sectoring was visually assessed. We then performed image processing using ImageJ to split channels and recolor the GFP channel.

## MCM assay

MCM assays were performed as previously published (*Maine et al., 1984*). However, samples were taken at only two time points. Therefore, we reported changes in frequency of 2μ plasmid rather than an estimated rate of loss per generation. This two-timepoint measurement also provided a more direct comparison to the SCAMPR assay. At time = 0 hr and 24 hr, cells were plated on both selective and non-selective media to determine what fraction of the population maintains the plasmid by virtue of encoding the selectable marker. Samples were plated at multiple dilutions to ensure between 30–300 CFU per plate. All strains containing the reporter plasmid were grown under G418 selection to ensure 2μ plasmid presence prior to the start of the assay. At time = 0 hr, cultures were transferred into liquid media with shaking, but without drug selection for 24 hr. After 24 hr, cultures were diluted in PBS and plated on YPD either with or without G418 selection at multiple dilutions, targeting 30–300 CFU per plate. Plates were incubated for 2 days, then colonies were manually counted to determine what fraction of the population were G418 positive. Calculations were based on whichever dilution gave a countable (30–300 CFU) plate. Multiple replicates (at least 8) were performed for each strain to measure variability in plasmid retention. A subpopulation of GFP-negative, G418-negative cells can be found even under selection. This 'phenotypic lag' occurs because of protein persistence following DNA loss; while cells that lose the plasmid die under selection, new

plasmid-free cells are constantly generated as well. We therefore normalize data to account for different starting frequencies of plasmid-negative cells by comparing cells grown with or without G418 selection for 24 hr relative to their starting frequency (*Figure 1B*).

## SCAMPR

SCAMPR samples were prepared as for MCM assays. When grown in 96-well format at 30°C, cultures were shaken using a Union Scientific VibraTranslator to ensure aeration. Fluorescence was directly measured by flow cytometry at 0 and 24 hr timepoints. A BD Canto-2 cytometer was used to collect cell data. FlowJo software was used for subsequent data analysis: samples were gated for single cells, omitting doublets/multiple cell clumps and any cell debris. Single cells were gated for GFP-positive and -negative populations, using GFP-negative strains and single-copy integrated GFP-positive strains as gating controls. Summary statistics (frequency of GFP-positive and -negative cells, GFP intensity) were exported from FloJo. Each strain was measured in at least triplicate per assay and means are reported here.

## Statistical analyses

For SCAMPR and MCM assay results, we determined significance by non-parametric tests. To compare two strains we used two-tailed Mann-Whitney tests, and to compare three or more strains, we used Kruskal-Wallis with Dunn's multiple comparison tests. Graphs were prepared and statistical analysis done using GraphPad Prism seven software.

## Screening for endogenous 2μ plasmid in natural isolates of *S. cerevisiae*

Natural isolates (*Supplementary file 1*) were generously shared by Dr. Justin Fay. DNA from these strains was isolated using a standard Hoffman and Winston preparation method, then probed by PCR and Southern blot (*Amberg et al., 2005*). Two pairs of primers were designed to amplify either *REP1* or *FLP1* (FLP1_F: CCACAATTTGGTATATTATG, FLP1_R: CTTTCACCCTCACTTAG, REP1_F: AATGGCGAGAGACT, REP1_R: CGTGAGAATGAATTTAGTA), the two best conserved coding regions of the plasmid, as previously described (*Xiao et al., 1991a*). Only strains that showed negative PCR results for both sets of primers were further validated by chemiluminescent Southern blot using the Thermo North2South kit (17097). Briefly, whole genome DNA was digested, run on an agarose gel in TAE, transferred to membrane and probed with chemiluminescent probes created from digested endogenous 2μ plasmid collected from BY4741 by Zymoresearch yeast plasmid miniprep kit (D2004). Gels and blots were imaged on a Bio-Rad ChemiDoc.

## Illumina sequencing, library preparation, and QTL mapping via bulk segregant analysis

We prepared high quality genomic DNA for sequencing using Zymoresearch Yeastar kits according to the manufacturer's instructions (D2002 - using chloroform method). Sequencing libraries were prepared using the TruSeq method for genomic DNA (Illumina), multiplexed and run on an Illumina HiSeq by the Fred Hutchinson Sequencing core facility to generate 50 bp paired-end sequences (SRA accession PRJNA637093). 100 bp paired-end reads for the lab strain, BY4742, were downloaded from the SRA database (accession SRR1569895). Reads that failed Illumina's 'chastity filter' were removed using a custom R script, and adapters and low-quality regions were trimmed using cutadapt with parameters -q 10 `--minimum-length` 20 (*Martin, 2011*). Trimmed read pairs were aligned to the sacCer3 reference genome assembly using BWA-backtrack (*Li and Durbin, 2009*). Mean coverage in non-overlapping 20 kb windows across the genome was calculated and plotted using R and Bioconductor.

For bulk segregant analysis, we first identified a conservative set of 47,173 high quality SNPs that distinguish the cross parents (Y9 and BY4742) as follows. Before SNP-calling, BWA output files were processed using Picard's MarkDuplicates tool and indels were realigned using GATK's RealignerTargetCreator and IndelRealigner tools (*Picard, 2020*; *DePristo et al., 2011*). We then called SNPs using samtools mpileup (parameters `--skip-indels` -t DP -uBg -d 6660) and bcftools call (parameters -vmO z -o), and counted reads matching each allele using GATK's VariantAnnotator DepthPerAlleleBySample module (with --downsampling_type NONE option) (*Li, 2011*). We used R and Bioconductor to further filter SNPs to obtain the final set of 47,173 SNPs, removing any that

overlapped repetitive elements, SNPs with QUAL score <200, SNPs with unusual coverage in any sample, and SNPs with an apparent mix of alleles in either of the haploid parental strains. We then ran MULTIPOOL in 'contrast' mode on allele frequencies at each SNP in the permissive and non-permissive pools to generate LOD scores across all chromosomes (*Edwards and Gifford, 2012*).

To identify candidate functional polymorphisms in each sequenced strain, we took two approaches: (a) we performed more sensitive SNP-calling, including small insertions and deletions; (b) to detect larger insertion/deletion events, we generated de novo assemblies from each strain, aligned them to the reference genome assembly, and identified locations where assemblies differed. In more detail, the first approach used processed alignments (see above) as input to GATK's HaplotypeCaller (parameters -stand_call_conf 30.0 -stand_emit_conf 10.0) (*DePristo et al., 2011*). Functional consequences of each variant were annotated using Ensembl's Variant Effect Predictor (*McLaren et al., 2016*). For the second approach (de novo assemblies), we performed error correction on the adapter-trimmed reads using musket (parameters -k 28 536870912) and then used SOAPdenovo2 across a range of k-mer sizes and fragment sizes, choosing the combination for each sample that yielded the assembly with highest N50 length as determined using QUAST (*Liu et al., 2013*; *Luo et al., 2012*; *Gurevich et al., 2013*). These Whole Genome Shotgun projects have been deposited at DDBJ/ENA/GenBank under the accessions JABVXK000000000, JABVXL000000000, JABVXM000000000, JABVXN000000000, JABVXO000000000 and JABVXP000000000. The versions described in this paper are versions JABVXK010000000, etc. We obtained tiling path alignments of each assembly to the sacCer3 reference genome assembly using MUMMER (nucmer parameters -maxmatch -l 100 c 500, delta-filter options -m) (*Kurtz et al., 2004*). Structural variants were determined from genome alignments using Assemblytics (variant size range 1 bp-100kb) (*Nattestad and Schatz, 2016*).

We identified all missense polymorphisms in the chrV 90% credible interval where genotype was shared between the non-permissive Y9 and Y12 strains, but distinct from the closely-related plasmid-permissive strain, UC5, and the permissive laboratory strain, BY4742 (*Supplementary file 3*). In addition, we expanded our analysis to include all candidate regulatory and synonymous polymorphisms at the *MMS21* locus (*Figure 4—figure supplement 4*).

## Structure visualization

The Cn3D viewer was used to visualize Thr69Ile on a crystal structure of *MMS21* with *SMC5* made available by *Duan et al., 2009*; *Wang et al., 2000*.

## Analysis of MMS21 natural variation

To examine natural variation in *MMS21* across *S. cerevisiae* strains and in other fungal species, we first extracted the *MMS21* (YEL019C) open reading frame from the reference assembly (sacCer3, chrV:120,498–121,301, - strand) and translated that sequence. We then used this *MMS21* protein sequence as the query in tblastn searches against various databases (*Altschul et al., 1997*). Searching the NR database, using taxonomic restrictions as needed, yielded *MMS21* sequences from *S. paradoxus* (XM_033909904.1), *S. eubayanus* (XM_018364578.1), *S. jurei* (LT986468.1, bases 125,344–126,147, - strand), *S. kudriavzevii* (LR215939.1, bases 100238–101041, - strand), *N. castellii* (XM_003677586.1) and *N. dairenensis* (XM_003671024.2). For additional orthologs, we downloaded individual genome assemblies from NCBI and performed local tblastn searches for *S. arboricola* (GCA_000292725.1; *MMS21* at CM001567.1:97,893–98,699, - strand), *S. mikatae* (GCA_000166975.1; *MMS21* at AABZ01000034.1:31,665–32,468, - strand) and *S. uvarum* (GCA_002242645, *MMS21* at NOWY01000012.1:107,550–108,353, + strand). Additional *S. cerevisiae* strain sequences come from our own de novo assemblies, where we used blastn to identify *MMS21*.

*MMS21* genotypes in the 1011 isolates previously sequenced (*Peter et al., 2018*) were accessed via that publication's supplementary data file 1011Matrix.gvcf.gz. Plasmid status was obtained from another supplementary file (*Supplementary file 1*) and cross-referenced with genotype in R.

## Acknowledgements

We thank past and present members of the Malik, Dudley and Raghuraman/Brewer labs for helpful discussions throughout this project. We especially thank Aimée Dudley and Mosur Raghuraman for

their constant support throughout this project. We also thank Yu-Ying (Phoebe) Hsieh, Tera Levin, Antoine Molaro, Courtney Schroeder, and Gavin Sherlock for their comments on the manuscript, and Howard Chang for reminding us of the Smc5/6 restriction of Hepatitis B viruses. We are grateful to Justin Fay for the generous gift of natural yeast isolates. We appreciate all the generous assistance and advice from Flow Cytometry and Genomics shared resource facilities at the Fred Hutchinson Cancer Research Center. This work was supported by an NSF graduate research fellowship (Grant No. DGE-1256082 to M.H.), NIH/NHGRI Genome Training Grant at the University of Washington (5T32HG000035-20 to M.H.), NIH R01 grant GM074108 (to H.S.M.) and an Investigator award from HHMI (to H.S.M.). The funders had no role in study design, data collection and analysis, decision to publish, or preparation of the manuscript.

## Additional information

### Funding

| Funder | Grant reference number | Author |
| --- | --- | --- |
| National Institute of General Medical Sciences | R01 GM074108 | Harmit S Malik |
| National Science Foundation | DGE-1256082 | Michelle Hays |
| National Human Genome Research Institute | 5T32HG000035-20 | Michelle Hays |
| Howard Hughes Medical Institute | Investigator award | Harmit S Malik |

The funders had no role in study design, data collection and interpretation, or the decision to submit the work for publication.

### Author contributions

Michelle Hays, Conceptualization, Data curation, Formal analysis, Supervision, Funding acquisition, Validation, Investigation, Visualization, Methodology, Writing - original draft, Project administration, Writing - review and editing; Janet M Young, Data curation, Software, Formal analysis, Validation, Investigation, Visualization, Methodology, Writing - original draft, Writing - review and editing; Paula F Levan, Investigation, Methodology, Writing - review and editing; Harmit S Malik, Conceptualization, Resources, Formal analysis, Supervision, Funding acquisition, Visualization, Writing - original draft, Project administration, Writing - review and editing

### Author ORCIDs

Michelle Hays https://orcid.org/0000-0002-8540-3516
Janet M Young http://orcid.org/0000-0001-8220-8427
Harmit S Malik https://orcid.org/0000-0001-6005-0016

### Decision letter and Author response

Decision letter https://doi.org/10.7554/eLife.62337.sa1
Author response https://doi.org/10.7554/eLife.62337.sa2

## Additional files

### Supplementary files

• Supplementary file 1. Natural *S. cerevisiae* isolates screened for the presence or absence of endogenous 2µ plasmids.

• Supplementary file 2. Engineered *S. cerevisiae* strains used in this study.

• Supplementary file 3. Missense polymorphisms in the 90% credible QTL interval for plasmid instability. The table lists missense polymorphisms shared between the non-permissive Y9 and Y12 strains, but distinct from the closely-related plasmid-permissive strain, UC5, and the permissive laboratory strain, BY4742.

• Supplementary file 4. Missense polymorphisms in Smc5/6 complex members and other SUMO ligases. All non-synonymous differences between Y9 and BY4742 strains in all components of the Smc5/6 complex (Smc5, Smc6, Nse1-6; Nse2 is a synonym of Mms21) and in SUMO ligases Siz1 and Siz2.

• Transparent reporting form

## Data availability

Raw sequencing data have been deposited to the SRA database, accession PRJNA637093. De novo assemblies are in GenBank with accessions JABVXK000000000, JABVXL000000000, JABVXM000000000, JABVXN000000000, JABVXO000000000 and JABVXP000000000.

The following datasets were generated:

| Author(s) | Year | Dataset title | Dataset URL | Database and Identifier |
|---|---|---|---|---|
| Hays M, Young JM, Levan PF, Malik HS | 2020 | Natural variation among *Saccharomyces cerevisiae* strains in resistance to 2-micron plasmid | http://www.ncbi.nlm.nih.gov/bioproject/?term=PRJNA637093 | NCBI BioProject, PRJNA637093 |
| Hays M, Young JM, Levan PF, Malik HS | 2020 | *Saccharomyces cerevisiae* strain Y9_Hap1, whole genome shotgun sequencing project | https://www.ncbi.nlm.nih.gov/nuccore/JABVXK000000000 | NCBI Nucleotide, JABVXK000000000 |
| Hays M, Young JM, Levan PF, Malik HS | 2020 | *Saccharomyces cerevisiae* strain Y12, whole genome shotgun sequencing project | https://www.ncbi.nlm.nih.gov/nuccore/JABVXL000000000 | NCBI Nucleotide, JABVXL000000000 |
| Hays M, Young JM, Levan PF, Malik HS | 2020 | *Saccharomyces cerevisiae* strain NC-02, whole genome shotgun sequencing project | https://www.ncbi.nlm.nih.gov/nuccore/JABVXM000000000 | NCBI Nucleotide, JABVXM000000000 |
| Hays M, Young JM, Levan PF, Malik HS | 2020 | *Saccharomyces cerevisiae* strain YPS1009, whole genome shotgun sequencing project | https://www.ncbi.nlm.nih.gov/nuccore/JABVXN000000000 | NCBI Nucleotide, JABVXN000000000 |
| Hays M, Young JM, Levan PF, Malik HS | 2020 | *Saccharomyces cerevisiae* strain PW5, whole genome shotgun sequencing project | https://www.ncbi.nlm.nih.gov/nuccore/JABVXO000000000 | NCBI Nucleotide, JABVXO000000000 |
| Hays M, Young JM, Levan PF, Malik HS | 2020 | *Saccharomyces cerevisiae* strain UC5, whole genome shotgun sequencing project | https://www.ncbi.nlm.nih.gov/nuccore/JABVXP000000000 | NCBI Nucleotide, JABVXP000000000 |

The following previously published dataset was used:

| Author(s) | Year | Dataset title | Dataset URL | Database and Identifier |
|---|---|---|---|---|
| Song G, Stanford University | 2014 | *Saccharomyces cerevisiae* strain genomes commonly used in laboratories | https://www.ncbi.nlm.nih.gov/sra/SRR1569895 | NCBI Sequence Read Archive, SRR1569895 |

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
