## [Decision Letter]

[Editors' note: this paper was reviewed by Review Commons.]

**Acceptance summary:**

Parasitic genetic elements offer a fascinating framework for the study of evolution as they battle with their hosts. In this paper, the authors identify mutations in yeast that may tilt the balance in favour of the host by increasing the loss rate of a parasitic selfish plasmid. This study lays the foundation for a new study system to better understand the evolutionary arms race between genetic parasites and their hosts at the molecular level.

**Decision letter after peer review:**

Thank you for submitting your article "A natural variant of the essential host gene *MMS21* restricts the parasitic 2-micron plasmid in *Saccharomyces cerevisiae*" for consideration by *eLife*. Your article has been reviewed by three peer reviewers at Review Commons, and the evaluation has been overseen by a Reviewing Editor and Detlef Weigel as the Senior Editor.

In this manuscript, the authors examine the genetic basis for parasitic plasmid resistance in budding yeast. They develop a new method that allows to measure plasmid copy number and retention at the single cell level. They leverage this tool to map QTLs underlying variation in plasmid retention or stability among strains of yeast. They identify a locus that explains a significant fraction of the variation found among strains. The candidate locus is involved in biological processes that have been associated with plasmid transmission in previous works. The study is of interest from a technical and conceptual standpoint and will be of broad interest.

From reading the comments, responses and the manuscript in details, it seems that two comments remain to be resolved.

First, you mention that you could include in the manuscript the results shown in response to reviewer #1, comment 4. Since a similar comment was made by another reviewer, it would indeed be important to include this analysis in a revised submission. In addition, the figure as it is now in the rebuttal is difficult to read.

Second, reviewer #1 points out in their comment #6 that allele replacement would be important to confirm the role of the candidate amino acid substitution. I understand that the paper itself goes beyond the identification of a precise variant and that the current COVID-19 conditions limit your access to lab facilities. However, since this claim appears to be key for some of the downstream analyses on the 1001 genomes and on closely related species, and for the discussion about the mechanisms by which the plasmid is unstable in Y9, this appears to be a key experiment. One of the sub-section title states for instance that this single amino acid variant in *MMS21* contributes to mitotic instability of the plasmid in Y9.

Here are some observations that make the contribution of this amino acid substitution uncertain. First, the Y9 instability appears to be dominant as measured in the heterozygote whereas the addition of the Y9 allele in BY does not affect its phenotype. Second, on Figure 5C, I appreciate that the phenotype of Het-Y9 *MMS21* is different from the Het, but a relevant comparison here would be Het-Y9 *MMS21* versus Het-4742 *MMS21*. These two strains are the ones for which the only difference is the nature of the allele at the locus and not the number of alleles. The statistical test for this difference is not reported but it is likely not significant given what is shown on the figure. The results shown could for instance (and one could think of other dosage effects) be consistent with a difference in expression of the two alleles acting in cis, which would also alter the phenotypes depending on which allele is deleted. Since the analysis is focusing on missense variants, regulatory variants have not been considered so this cannot be ruled out. I therefore believe that allele replacement would be needed to confirm the identity of the causal variant.

---

## [Author Response]

We would like start by thanking all three reviewers for their time, effort and extremely helpful comments. This has been a very constructive review process and we are grateful.

Reviewer #1 (Evidence, reproducibility and clarity):The paper by Hays et al. addresses the evolution of host genomes to eliminate parasitic selfish genomes that coevolve with them. They use the multi-copy 2-micron yeast plasmid as a model selfish genome to address this issue. Using a series of cleverly designed experiments, the authors identify a high confidence quantitative trait locus (QTL) on chromosome V in the Y9 yeast strain associated with mitotic plasmid instability. Furthermore, they demonstrate Thr69Ile substitution in MMS21 (a SUMO E3 ligase) to be a heritable dominant determinant for plasmid exclusion. The work is significant for its conceptual and methodological advances.Although the multigenic nature of the commensal relationship between the plasmid and host would make the task rather arduous, the present study exemplifies the utility of natural variations in understanding the mechanisms of genetic conflicts and their resolution during the coevolution of selfish genetic elements and their hosts.

We thank the reviewer for their kind comments and constructive suggestions and corrections, which we have now incorporated into our revision as detailed below.

1) SCAMPR; Figure 1, Relevant text. The fluorescence based single cell high throughput analysis of plasmid retention, SCAMPR, is an important advancement. However, the use of reporter plasmids containing insertion of markers that minimally disturb native plasmid organization, and containing no bacterial vector sequences, has been described earlier.

We thank the reviewer for pointing out this oversight. We have now modified the main text to specify that our strategy was not the first to mark the 2μ plasmid without bacterial sequences.

2) Figure 1C and the relevant text. For total missegregation of the GFP-plasmids, the mother cell should contain 2 x n plasmid copies, and therefore twice the GFP intensity. The schematic diagram in Figure 1C does not seem to depict this increase in copy number in plasmid retaining cells. The legend suggests that “no change” in the median GFP intensity in plasmid containing cells is expected. And according to the text, the copy number is not too high to saturate the signal detector. Perhaps some clarification here might help. Is there potential suppression of plasmid replication at high copy numbers? Or do such cells have a growth disadvantage and are selected against?

The reviewer raises an important question and one that we tried to address in our main text but have revised for clarity: we do not see a population of super-green cells. We speculate that this is because these cells are not fit. Lack of plasmid segregation is likely to result in cumulatively higher levels of plasmid retention in “mother” cells as the reviewer points out, and others have shown that high copy numbers of the plasmid are detrimental to the host. However, we agree with the reviewer that the SCAMPR schematic can make clear that this may be why we do not see the super-green cells that we would expect to see under a segregation defect. We have edited both the main text and amended the figure to reflect this important point, which was also raised by reviewer #2. Thank you for helping us to improve the clarity of this point.

3) Related to point 2 above. One wonders whether the GFP-plasmid is capable of Flp mediated amplification and copy number correction? This aspect has not received scrutiny, understandably because of the emphasis of this study on “plasmid resistance/exclusion”. In principle, there is nothing to preclude plasmid amplification by the generally accepted Futcher-Volkert-Broach mechanism. Copy number correction in the case of unequal segregation might be responsible for the lack of shift in median GFP intensity. One is curious to know whether a plasmid harboring inactive FLP or a nonfunctional FRT target site would exhibit a different behavior.

This is an excellent point. We believe that the modified reporter plasmid should be fully capable of FLP copy number repair because of the insertion location (we now make this point more explicitly when we cite the previous reference about our choice of insertion site). Although we do not test this possibility directly, we expect a FLP minus plasmid would be less stable in both the lab strain (as others have observed) and Y9 cells. Since we wished our reporter to be as similar as possible to the native 2μ plasmid (to focus on host interference with normal plasmid function), we did not study mutations in Flp or FRT sites, but this is an exciting research avenue to pursue in the future.

4) Figure 2. From panel A of the figure, plasmid-free colonies appear to be larger than plasmid containing colonies – more so for the Y9, Y12 and YPS1009 strains than BY4742. The phenotype would be consistent with a growth disadvantage of plasmid containing cells in the first three strains. The effect does not appear to be caused by GFP expression/toxicity as the size difference is less apparent for BY4742.

We thank the reviewer for this observation, which was also commented on by reviewer #2. Indeed, as the reviewer suggests, this could suggest an increase in plasmid fitness cost in the wild strains’ backgrounds. Since it can be difficult to decouple 2μ plasmid instability from a growth advantage of plasmid-lacking strains, we performed growth curve assays of Y9 and BY4742 haploid strains either lacking or containing plasmids

(with G418 selection to maintain plasmids). We found that the growth curves from plasmid-lacking BY4742 strains were indistinguishable from Y9 strains. Moreover, both strains had lower fitness in the presence of 2m plasmids (and G418 selection) but this fitness drop was also indistinguishable during log phase growth. This result argues against a strong Y9-specific plasmid carriage cost. If recommended by the editor, we would be happy to include this result in our revision.

Of course, it is possible that there is a more subtle relative fitness cost that can only be measured under competitive co-culture conditions, but this would require significant re-engineering of strains. This is why we leave open this possibility in the revised Discussion.

5) Text related to Figure 3—figure supplement 1. If plasmid segregation is strongly coupled to chromosome segregation, in accordance with the hitchhiking model, it would be quite unlikely that plasmid exclusion would be a monogenic effect. I think the present results are consistent with the interaction of MMS21 with the Smc5/6 complex, which appears to have pleiotropic effects on chromosome dynamics and function.

We agree with the reviewer; there are many possible genes that could pleiotropically affect both host chromosome stability and plasmid stability.

6) Figure 5A and related text. Is it possible to replace the MMS21(Thr69) in BY4742 with MMS21(Ile69) from Y9? Is that substitution compatible with normal chromosome functions? The expectation based on the hemizygous diploid experiments is that the Y9 allele would destabilize the GFP-plasmid. Based on the results from Figure 5B, either MMS21(Thr69) or MMS21(Ile69) is functional in the diploid background.

The reviewer is right on several points. Our hemizygous *MMS21* experiment in Y9/BY4742 heterozygous diploids demonstrates that either MMS21 (Thr69 or Ile69) is functional in the het. diploid background, at least in terms of providing *MMS21* essential function. Although we previously showed that addition of the Y9 *MMS21* allele into an intact BY4742 background does not appear sufficient to significantly lower plasmid stability (Figure 5A), it is possible that the presence of the BY4742 *MMS21* allele masked the Y9 allele’s plasmid instability phenotype. We can create a Y9 *MMS21*-only version of a BY4742 strain to assess viability differences.

We would also like to measure plasmid stability in this allele swap strain. However, due to COVID-related restrictions placed on our FACS facilities, SCAMPR is not possible at this time. The MCM assay is possible; however, the plasmid stability phenotype difference in the hemizygous mutants is subtle, even by SCAMPR (which samples tens of thousands of cells per assay rather than tens of colonies per plate). We therefore anticipate that this phenotypic difference would be overwhelmed by noise in the MCM assay. However, we would undertake these experiments if they are required by the editors.

7) The final paragraph of Results before Discussion. Based on the location of Thr/Ileu in MMS21, the proposed functional relevance of the MMS21 variants via Smc5/6 interaction would seem reasonable. Are there variations in the Smc5/6 complexes of BY4742 and Y9? And are the MMS21 variants able to functionally interact with both Smc5/6 complexes? This issue would be related to the point raised under 6 above.

In response to the reviewer query, we have now carried out a comprehensive analysis of variants in all members of the Smc5/6 complex which we now report in our revision (Supplementary file 4). However, we detect no polymorphism in the Smc5 sites that would likely interact with residue 69 of Mms21. Beyond these inferences, we are not able to make stronger statements about any epistatic interactions between *MMS21* and any other components of the Smc5/6 complex in the absence of much more detailed biochemical analyses.

8) The SUMO E3 ligase activity of MMS21. It is implied in the text that the ligase activity of MMS21 is not involved in plasmid plasmid exclusion/retention. Has this been tested by inactivating mutations. To my knowledge, most of the published effects of MMS21 have been interpreted in terms of its E3 ligase activity in the context of the Smc5/6 complex. An E3 ligase independent function for MMS21 in commensalism would be quite interesting.

We think the reviewer is referring to our hypothesis that the Mms21 polymorphism is more likely to affect its interaction with Smc5/6 rather than the SUMO ring domain. We are excited by the prospect of follow-up studies to understand the mechanism of this polymorphism on Smc5/6 function. We feel these mechanistic experiments are outside of the scope of this current paper. We do more explicitly point out this possibility in our revised Discussion: “Interestingly, a recent study demonstrated that even the SUMO ligase function of Mms21 depends on its docking with the Smc5/6 complex. Thus, the polymorphism in the Mms21-Smc5 interaction site could either affect Mms21’s SUMO ligase functions or other functions of the Smc5/6 complex.”

If the editors require, we could explore plasmid stability in other *MMS21* mutants, known to be deficient in sumoylation in others’ hands. However, there are several technical concerns we have about undertaking this experiment. *MMS21* is an essential gene, so deleterious mutations in this gene are likely to cause substantial host fitness defects, which would confound any observed changes in plasmid stability. We again note that we are currently unable to perform SCAMPR due to COVID restrictions, so while we could follow this experiment up by MCM analysis, the sensitivity and sample size limitations of that assay (in conjunction with host viability concerns) are likely to make the results of this experiment challenging to interpret.

9) Is MMS21 function always coupled to Smc5/6? Although the location of the Thr69Ile location is consistent with MMS21 interaction with Smc5/6, there is no direct evidence to suggest that this interaction is involved in the plasmid stability/instability phenotype. Is there any known instance where MMS21 functions independently of Smc5/6?

We are not aware of any studies that have discovered non-SMC5/6 roles for Mms21. In contrast, a recent paper quite conclusively demonstrated that even the SUMO ligase functions of Mms21 are tied to its membership in the Smc5/6 complex (we now cite this paper more explicitly to make this point). Thus, the location of the polymorphism at the Smc5 interaction site is most consistent with Mms21's interaction with Smc5/6 function being affected.

Reviewer #1 (Significance):The paper addresses the genetic determinants that influence the coevolution of host genomes and associated selfish DNA genomes by exploiting natural variations occurring in the host species. The problem and the approaches are of broad significance and interest.

We are grateful for the positive appraisal of the reviewer and their constructive suggestions to improve our paper.

Referees cross commenting :The points raised in the other reviews are valid.The issue that I am struggling with is the schematics in Figure 2 and the lack of change in median fluorescent intensity. The only way to reconcile the apparent discrepancy is to assume that cells with the high plasmid copy are eliminated from the population (selected against?). Nevertheless, if plasmid replication is not affected, in a daughter pair with one receiving all plasmid copies should be 2 n in copy number, which is not what the figure shows.If I am missing something, could one of the other two reviewers correct me. please?

As we highlight in point 2 (above), the reviewer has raised an important issue that we previously only addressed in the text. We now expand on this possibility in the main text, have included this inference in Figure 1C, and explicitly state our hypothesis that cells with high plasmid copy number (super-green) are most likely not observed due to proliferation defects.

Reviewer #2 (Evidence, reproducibility and clarity):Summary:The 2-micron plasmid is a selfish DNA element, typically found at high copy number in most lab strains of the yeast Saccharomyces cerevisiae. In this paper, a PCR survey of 52 natural isolates identified 3 strains lacking 2-micron plasmid. A novel single-cell high-throughput plasmid retention assay (SCAMPR) was developed and in combination with traditional yeast genetics, used to demonstrate plasmid instability in one of these was due to an inherited, dominant, multigenic trait. Next generation sequencing of pooled haploid segregants (identified as either permissive or non-permissive for plasmid maintenance using SCAMPR) allowed quantitative trait locus (QTL) mapping. A focus on candidate genes in the major QTL identified a single amino acid change (Thr69Ile) in a conserved essential SUMO E3 ligase, Mms21, as being associated with reduced 2-micron plasmid stability. Variation in Mms21 in species closely related to *S. cerevisiae* and correlation of the Ile69 allele with lower likelihood of possessing A-type 2-micron plasmids in *S. cerevisiae* was also reported. Mms21 is a subunit of the conserved Smc5/6 complex which plays an essential, but incompletely understood role in chromosome maintenance. Prior crystal structure data was used to identify the position of Thr69Ile in Mms21 as being in contact with the Smc5 subunit rather than in the SUMO ligase domain.The paper is clearly written, and the key conclusions are well supported by the data provided. The data and methods are generally clearly presented and represent a useful template for others undertaking similar studies. Replicates and statistical analysis are adequate.

We are very grateful for the positive summary and constructive feedback from the reviewer.

Comments:1) On page 9, last line in 2nd paragraph in this section – the consequences of plasmid missegregation here are correctly stated as "some cells inheriting no plasmid, while others maintain or even increase plasmid copy number (Figure 1C)". However, in Figure 1C, the accompanying diagram is misleading as it shows copy number staying uniform in the cells that have plasmid when it would be expected to increase in these due to plasmid replication having doubled the copy number and lack of delivery to daughter cells due to missegregation. Similarly, the legend for this part of Figure 1 states that missegregation would "cause a rapid increase in GFP-negative cells but no change in the median expression of GFP-positive cells". The prediction should be no change or an increase in the median expression for the GFP-positive cells as stated. What is striking in Figure 1—figure supplement 1B is that the predicted increase it not seen. This suggests that for the Y9 strain, cells that received the higher than normal number of plasmids are very rapidly lost from the population. This would be consistent with the high number being toxic. Figure 1C and the legend should be amended to reflect the predictions and also to make it clearer that the plasmids are not lost from the cells, rather they are being lost from the cell population, due to failure to be delivered to daughter cells and presumably because plasmid-free cells rapidly out-compete plasmid-containing cells during the 24 hr when they are cultured in the absence of selection.

We completely agree with the reviewer, and appreciate the assistance in clarifying this point. This concern was also raised by reviewer #1 (see point 2) and we have revised the text and figure to incorporate these comments. We thank both the reviewers for pointing this inconsistency to us and suggesting remedies to improve its clarity and accuracy.

2) The deletion of the URA3 gene in BY4742 is referred to as a large indel. I appreciate that indel is the term used for genomic comparisons where the process leading to the difference is unknown but since we know that the URA3 gene was actively deleted in the construction of BY4742, this seems like a misleading term here.

We agree with the reviewer and have edited the language in our revision to reflect this.

3a) One part that would benefit from a bit more clarity is on page 17, where the assumption is made that since nibbled colonies were not observed when their GFP-tagged 2-micron plasmid was introduced into the Y9 cells, then any change to Mms 21function due to the Y9 allele is not synthetically lethal with high levels of the plasmid. I assume they mean the abnormally elevated plasmid copy number that occurs in sectors of ulp1 mutant colonies causing cells in those sectors to cease proliferating. A reader would be more apt to interpret the term "synthetic lethality" as being immediate inviability, or at least that the cells would not be able to form a viable colony, so this might not be the most helpful term here or at least requires clarification. For the Y9 strain, the combination of MMS21 allele and 2-micron can be a negative genetic interaction (synthetically sick) without giving rise to nibbled colonies.

We agree with this comment from the reviewer and have amended the text to be more precise with our description.

3b) There does seem to be a suggestion of this in Figure 2A where the Y9 colonies that express GFP seem on average to be smaller than those that do not. This contrasts with BY4742 where GFP-positive colonies similar in size to large GFP-negative ones are seen. It might be useful to compare the relative growth rates of Y9 versus BY4742, transformed with a 2-micron plasmid versus transformed with a CEN plasmid and versus both untransformed to see if there is a significant synthetic sick phenotype only when a 2-micron plasmid is present.

We thank the reviewer for this comment and observation. We note that reviewer #1 also made this observation (reviewer #1 point 4), which we now have addressed in our response to reviewer #1.

4) 2-micron plasmids are stated to "physically localize to the nuclear periphery" but the cited paper does not conclude this and other reports seem to be somewhat at odds as to localization, concluding that the plasmid is either spindle pole body-associated in mitotic cells (Mehta et al., 2005), spindle-associated (Velmurugan et al., 2000), explicitly not at the nuclear periphery (ScottDrew et al., 2002), and at the nuclear periphery, but only in meiosis (Sau et al., 2014). Under the circumstances, this statement needs to be removed or qualified.

We thank the reviewer for pointing out our erroneous phrasing, which we now removed and edited this section of the manuscript accordingly.

5) For Figure 3B, the legend and should be rewritten to make it clear whether these are ~600 random spores from each of the homozygous Y9 and BY4742 diploids and therefore directly comparable to the ~600 random spores from the heterozygotes in Figure 3C or if these are the distributions from the respective parental haploids.

We thank the reviewer for pointing out the lack of clarity here. We now clarify that these represent parental haploid distributions from across experiments (red and blue), and not homozygous progeny distributions.

Reviewer #2 (Significance):This study is one of the first to leverage yeast genetics and the extensive genomic resources now available for yeast to identify natural variation in yeast populations that contribute to a trait of interest. Specifically, here the authors identify a variant of the SUMO ligase Mms21 that is associated with reduced mitotic stability of the 2-micron plasmid, a selfish genetic element, that despite conferring a growth fitness defect, is found in many yeast strains. The results are significant due to:1) The conserved and essential role of the Smc5/6 complex, of which Mms21 is a subunit. Unlike the related cohesin and condensin complexes, the function of Smc5/6 complex is much less well understood.2) The value of identifying non-conditional alleles, such as the Mms21 Thr69Ile, by this approach that can be used as tools for dissecting function of essential genes where the lethality of knock outs precludes many conventional analyses.3) This is the first report to suggest a role for the Smc5/6 complex in 2-micron plasmid maintenance and is particularly intriguing in light of prior work implicating human Smc5/6 in limiting Hepatitis B virus.4) The SCAMPR assay developed for this study will facilitate future high-throughput analyses of plasmid maintenance.5) Taken together, the results represent a valuable addition to our understanding of 2-micron plasmid distribution and host interactions.This paper will be of interest to those who use yeast as a model organism for molecular genetics and cell biology studies, those in the biotechnology and industrial sector who work with vector systems for model and non-model yeasts, and researchers with an interest in evolutionary biology, population genetics, eukaryotic chromosome biology, DNA repair and cell cycle regulation.

We are highly grateful for the accurate and positive summary of our article by the reviewer and for their comments towards improving and correcting our paper.

Referees cross commenting:The other reviews seem quite reasonable to me. Clearly all three of us found this to be an interesting paper.You are not missing anything, the cartoon in Figure 1C is incorrect as I also indicated in my review. In the text, it does say that plasmid copy number could increase in some cells if the plasmid missegregates (they should have stated that it would increase if replication is not affected). If this is total lack of delivery to daughters, copy number would double in mother cells and if GFP intensity is not saturated, one would have expected an increase in fluorescence for those cells (the cartoon should show a shift to the right in in the median GFP intensity for the GFP-positive cells. They did not see this (Figure 1—figure supplement 1) which could be due to the higher copy number being toxic resulting in loss of those cells from the population during the 24 hr culture under non-selective conditions as plasmid-free cells out-compete plasmid-plus ones. It did look like most of the GFP signal was gone after 24 hrs for the Y9 strain. This would not be unexpected. The authors do need to fix the cartoon and at least speculate as to why they did not see the expected increase.

We agree with these comments and suggestions by both reviewers #1 and #2 and have modified the figure and main text accordingly in our revision.

I agree that they should also have mentioned Flp-mediated correction of copy number in this context as it likely contributes to maintaining the median level of GFP intensity in the population of GFP-positive cells (if at least some plasmid copies are delivered to daughters in the absence of effective equipartitioning), although Flp cannot reduce copy number in cells that have already received higher than normal numbers.

As reviewer #2 states, Flp-mediated correction is likely to increase, not decrease copy number, which is why we favor the hypothesis that the lack of observed super-green cells with increased plasmid copy number is most likely the result of loss of proliferative fitness of these cells. Nevertheless, we agree that the issue of Flp-mediated amplification needs to be brought up in the main text as well as our motivation for working with a Flp positive plasmid: because the endogenous plasmid is capable of copy number recovery in the lab strain, but is not natively found in Y9, we wondered if Y9 was capable of restricting endogenous plasmid propagation (even with this amplification system intact). Our reporter plasmid is therefore as similar to an endogenous plasmid as we possible while still facilitating plasmid selection and screening. We have tried to better address these Flp ideas in our manuscript revision, and greatly appreciate the reviewers’ suggestion.

Reviewer #3 (Evidence, reproducibility and clarity):In this manuscript, the authors examine diversity in 2-micron plasmid stability across Saccharomyces cerevisiae strains. The 2-micron plasmid is a selfish genetic element that has co-evolved with the Saccharomyces genus. Notably, they find that some *Saccharomyces cerevisiae* strains lack 2-micron plasmids. These strains also show low 2-micron plasmid stability after plasmid introduction. The authors map a major locus involved in this phenotype and localize its effect to MMS21, which had not been previously connected to 2-micron plasmid stability.This manuscript is written in a way that is easy to read. The authors do a good job of conveying their results in a manner that is interesting, but not hyperbolic. Technically, I didn’t have any qualms with the work; it is rigorous.

We thank the reviewer for their constructive comments and positive appraisal.

More specific comments follow:The first two paragraphs of the results are focused on assays that have been historically used to measure plasmid stability. This information is technical background. Should it be elsewhere, such as in the introduction or a supplementary note?

We appreciate the reviewer’s point. We did consider putting details about the earlier assays in the Introduction but felt that discussing them together with the SCAMPR assay in the Results made clear where our reporter plasmids and strategies differed and where it built upon previous assays. We would therefore prefer to leave it as it is in the beginning of the Results section if possible.

How much larger is the GFP-2 micron reporter plasmid than the endogenous 2 micron plasmid? Could difference in size between be another possible explanation for difference in stability?

Previous reports have shown that the 2μ plasmid can tolerate an insertion of ~3.9 kb. The cassette introduced in is 2703 bp. We now specify these additional details in the revision and emphasize the relevant citation.

I am not sure that the segregation data provide much clarity on polygenicity. The data don’t rule out the possibility that a good amount of the variability is just stochastic and the Castle-Wright estimator has limited utility. The authors might want to be more equivocal in this paragraph.

We agree with this comment. We have edited the text accordingly (we do, however, retain mention of the Castle-Wright estimate, as the expected complexity of the trait contributed to our choice of using a bulk segregant approach, and we are often asked for this estimate).

During discussion of confidence intervals, it might be worthwhile to note the position of the peak marker(s). Often those are pretty close to the causal gene in these bulk segregant studies.

We greatly appreciate this suggestion. Indeed, the T69I SNP is only 1.2kb away from the peak markers! We now include this information in the text, with details of peak locations and confidence interval coordinates in the figure and supplementary figure legends.

I found the emphasis on SCAMPR at some points distracting. I recognize the assay is an improvement upon other techniques, but the generalist reader might not find the assay that compelling.

Although we think this is a fair comment by the reviewer, we note that the SCAMPR assay allowed us the throughput, sensitivity and reproducibility to do the genetic mapping studies. Since this method was critical for our approach and is not published separately at this time, we feel most comfortable leaving SCAMPR as one focus of this paper.

If any additional work were to be done, I would make sure MMS21 is the sole quantitative trait gene at the detected locus. The locus was not resolved very precisely and, as the authors themselves note, other quantitative trait genes affecting the same trait might be present. I do not regard this work as essential.

We appreciate the comment that the mapping of additional resistance genes is a definite avenue for future work.

Reviewer #3 (Significance):Whether genetic differences among hosts impact the propagation of selfish genetic elements is an interesting question. This paper not only demonstrates that host genetics affect the stability of selfish genetic elements, but also provides insights into molecular mechanism. A number of additional factors likely also contribute, but this does not diminish the MMS21 finding. I thought this was a nice story.

We appreciate the kind comments by the reviewer, and their helpful suggestions to streamline and improve the paper.

[Editors' note: further revisions were suggested prior to acceptance, as described below.]

First, you mention that you could include in the manuscript the results shown in response to reviewer #1, comment 4. Since a similar comment was made by another reviewer, it would indeed be important to include this analysis in a revised submission. In addition, the figure as it is now in the rebuttal is difficult to read.

We agree, and now provide growth curves as an additional Figure 2—figure supplement 2, along with detailed methods.

Second, reviewer #1 points out in their comment 6 that allele replacement would be important to confirm the role of the candidate amino acid substitution. I understand that the paper itself goes beyond the identification of a precise variant and that the current COVID-19 conditions limit your access to lab facilities. However, since this claim appears to be key for some of the downstream analyses on the 1001 genomes and on closely related species, and for the discussion about the mechanisms by which the plasmid is unstable in Y9, this appears to be a key experiment. One of the sub-section title states for instance that this single amino acid variant in MMS21 contributes to mitotic instability of the plasmid in Y9.Here are some observations that make the contribution of this amino acid substitution uncertain. First, the Y9 instability appears to be dominant as measured in the heterozygote whereas the addition of the Y9 allele in BY does not affect its phenotype. Second, on Figure 5C, I appreciate that the phenotype of Het-Y9 MMS21 is different from the Het, but a relevant comparison here would be Het-Y9 MMS21 versus Het-4742 MMS21. These two strains are the ones for which the only difference is the nature of the allele at the locus and not the number of alleles. The statistical test for this difference is not reported but it is likely not significant given what is shown on the figure. The results shown could for instance (and one could think of other dosage effects) be consistent with a difference in expression of the two alleles acting in cis, which would also alter the phenotypes depending on which allele is deleted. Since the analysis is focusing on missense variants, regulatory variants have not been considered so this cannot be ruled out. I therefore believe that allele replacement would be needed to confirm the identity of the causal variant.

There are several excellent points made here, including some that were not addressed by us and the reviewers. We note each of them here and our proposed remedy, which we hope will prove satisfactory.

1) In Figure 5B, as you note, the phenotype of Het minus Y9 *MMS21* is different from the Het, and you are right that we did not compare the Het minus Y9 *MMS21* to the Het minus BY4742 *MMS21*, which would also correct for the allelic dosage comparison. Surprisingly, while we had previously tested the significance of this difference, somehow it did not make it into our manuscript. We gratefully acknowledge your attention to this detail and now include this comparison. Although this looks like a modest difference in the figure, it is highly significant (p=0.0013, Kruskal-Wallis test). Indeed, one of SCAMPR’s advantages is large sample numbers with relatively tight variance. We have amended Figure 5 accordingly. As you can see, this analysis demonstrates that with copy number held the same, removal of BY4742 versus Y9 *MMS21* has significantly different effects on plasmid stability. We apologize for the omission of this important statistical comparison in our earlier version.

2) Your second point is that we did not adequately address whether it is regulatory sequence changes or the missense change in Y9 *MMS21* that is responsible for the instability phenotype. We have now subjected *MMS21*’s regulatory regions to the same comparative genomics approach we used in our original submission to identify candidate coding SNPs. Our whole genome sequencing shows that in addition to the *MMS21* missense SNP, Y9 differs from BY4742 at 4 synonymous sites within the ORF and a total of 11 sites in the two flanking (candidate regulatory) regions. For all 16 of these SNPs, the Y9 allele is shared with the closely-related plasmid-lacking sister strain Y12. We next examined the close outgroup UC5, which still bears plasmids according to our PCR survey: while the *MMS21* missense SNP and two synonymous SNPs differ between Y9 and UC5, and thus strictly correlate with the plasmid instability phenotype, the candidate regulatory SNPs do not. We have added this analysis (new Figure 5—figure supplement 1) to our revision to explain why we favor the missense mutation rather than regulatory sequences as causal for plasmid instability. Despite this reasoning, we also acknowledge in our Discussion the possibility that regulatory sequences might play a role.

The ideal disambiguation experiment between regulatory and coding SNPs would be allele swaps in the Y9/BY Het background. However, this experiment is not trivial, because the Y9 genome is much less amenable to engineering than standard lab strain genomes. Engineering allelic loss of *MMS21* in the Het background took us nearly two years under ideal (pre-COVID) circumstances, so we are unable to do this experiment in a reasonable time frame. We hope that your concerns will be sufficiently addressed by the addition of our analysis of regulatory and coding differences (above) and the statistical significance of plasmid instability differences between reciprocal deletions in the Het background (above).

3) We would also like to clarify why we do not expect the Y9 *MMS21* allele introduced into BY4742 (Figure 5A) to be sufficient to drive plasmid instability in the BY background. Our tetrad analysis (Figure 3—figure supplement 1), the phenotype distribution among F1 progeny (Figure 3C) and our QTL analysis (Figure 4B) strongly suggest plasmid instability is a multilocus trait. We believe that the combination of Y9 *MMS21* together with other genetic variants in the Y9 genome is required to explain the full plasmid instability phenotype. All of our *MMS21* genetic data are consistent with a multilocus dominant trait: first, while we do observe loss of instability upon removal of Y9 *MMS21* but not BY4742 *MMS21* from the heterozygote, this loss only partially recapitulates the parental phenotypes; second, addition of Y9 *MMS21* (with the Y9 native regulatory sequence) to the BY background is not sufficient to confer plasmid instability alone.